# Morphological and molecular evidence for first records and range extension of the Japanese seahorse, *Hippocampus mohnikei* (Bleeker 1853) in a bay-estuarine system of Goa, central west coast of India

**Sushant V. Sanaye**[1], **Rakhee Khandeparker**[1], **Anantha Sreepada Rayadurga** [1]*, **Mamatha S. Shivaramu**[1¤], **Harshada Kankonkar**[1], **Jayu Narvekar**[2], **Mukund Gauthankar**[1]

**1** Aquaculture Laboratory, Biological Oceanography Division, CSIR-National Institute of Oceanography, Dona Paula, Goa, India, **2** Physical Oceanography Division, CSIR-National Institute of Oceanography, Dona Paula, Goa, India

¤ Current address: CSIR-Central Food Technological Research Institute, Mysuru, Karnataka, India
* sreepada@nio.org

## Abstract

Accurate information of taxonomy and geographic range of seahorse species (genus *Hippocampus*) is the first step in preparing threat assessments and designing effective conservation measures. Here, we report first records and a range extension of the Japanese seahorse, *Hippocampus mohnikei* (Bleeker, 1853) from the Mandovi estuarine ecosystem of Goa, central west coast of India (CWCI) based on morphological and molecular analyses. The morphometric and meristic traits, particularly short snout (29–35% head length), double cheek spine, low coronet, long tail (51.2–57.9% of standard length), 11 trunk rings, 37–39 tail rings, 15–16 dorsal and 12–14 fin rays observed in four collected specimens matched with the reported key diagnostic morphological criteria of vouchered specimens of *H. mohnikei*. The seahorse mitochondrial cytochrome oxidase subunit I (COI) and cytochrome *b* (Cyt *b*) genes were partially sequenced for conclusive genetic identification of the species under study. Molecular analysis showed that all four individuals clustered together suggesting a monophyletic lineage. Using the maximum similarity with GenBank database, maximum likelihood network and subsequent morphological analysis, the identity of the collected seahorse species was reconfirmed as *H. mohnikei*. With this new report, the geographic range of *H. mohnikei* extended significantly to the west from its previously known range. This new sighting of *H. mohnikei* could indicate a long-distance dispersal facilitated by the prevailing oceanic circulation in the Indo-Pacific region or increased habitat suitability in bay-estuarine systems of Goa, CWCI. Comparison of the pair-wise genetic distances (Kimura 2-parameter) based on COI and Cyt *b* sequences revealed that the specimens examined in this study are genetically closer to *H. mohnikei* populations from Vietnam and Thailand than they are to those in Japan and China. To test the hypothesis whether *H. mohnikei* are vagrants or previously unreported established population, long-term inter-annual sampling and analyses are warranted.

**Data Availability Statement:** All relevant data are within the paper and its Supporting Information files.

**Funding:** The author(s) received no specific funding for this work.

**Competing interests:** The authors have declared that no competing interests exist.

## Introduction

Seahorses (Syngnathiformes; Syngnathidae) belonging to the single genus, *Hippocampus* (Rafinesque, 1810) are a fascinating and remarkable group of fishes with their unusual body shape and their biology, with males incubating the fertilized eggs in a brood pouch [1]. They are small, cryptic, and sedentary marine fishes that occur worldwide in shallow temperate to tropical waters in a wide variety of habitats, including seagrass beds, estuaries, coral and rocky reefs and mangroves [2–4]. Their remarkable ability to camouflage with structurally complex habitats further reduces the risk of predation [5], while providing stealth for prey capture [6]. However, these biological traits often also make them challenging for scientists to research and quantify.

The primary step in preparing threat assessments and designing effective conservation of seahorse populations is highly dependent on the precise identification of individual species in and beyond known geographic distributions [7]. Out of 44 recognized seahorse species (Genus *Hippocampus*) throughout the world [8], *H. kelloggi*, *H. kuda*, *H. spinosissimus* and *H. trimaculatus* are common in Indian coastal marine waters [9,10]. In addition, sightings based on morphological identification of single specimens of *H. mohnikei* from Palk Bay, southeastern India [11], *H. borboniensis* [12] and *H. montebelloensis* [13] from the Gulf of Mannar, southeast coast of India and *H. camelopardalis* from the Gulf of Kachchh, northwest coast of India [14] have been reported. Another seahorse species, *H. histrix*, is also suspected to occur in Indian coastal waters [15]. The validity of *H. borboniensis* and *H. montebelloensis* as separate species, however, is currently under revision due to their synonymity with *H. kuda* and *H. zebra*, respectively [16].

The utility of morphological traits commonly used in diagnosis of *Hippocampus* spp. is quite challenging as they lack certain key physical features (e.g. pelvic and caudal fins), and have high variability in body proportion, color (camouflage) and skin filaments, fin-ray and trunk-ring numbers; any of these traits might overlap between species [3,17]. On the other hand, molecular methods have proved helpful in resolving morphologically challenging seahorse taxonomy [16,18], phylogenetic relationships within the genus *Hippocampus* [19,20], natural species boundaries [21–23] and genetic variability [22,24] of many seahorse species. An integrated approach combining morphological and genetic analyses [25] would aid in the management of demographically separate populations as independent units and allow international legal mechanisms and international agreements such as the Convention on International Trade in Endangered Species (CITES) to work effectively [26].

Generally, seahorses are sedentary fishes with typically small home ranges and patchy distribution in sheltered areas such as seagrasses and seaweeds [2,27,28]. However, instances of limited migrations by seahorses in search of proper habitat, food or holdfasts have been reported [28,29]. Evidences of long-distance dispersal of seahorse species via rafting (floating debris and seaweeds) [2,12,30] and subsequent colonization [21] have also been documented. The role of oceanic currents facilitating long-distance dispersal and range expansion of many seahorse species has also been highlighted [31–36].

The Japanese or Lemur-tail seahorse, *Hippocampus mohnikei* (Bleeker, 1853) is a small coastal seahorse (5–8 cm) inhabiting seagrasses, mangroves, oyster beds and mussel farms [37]. Original records indicated that the distribution of *H. mohnikei* is restricted or native to the coastal waters around Japan and Vietnam [17, 38–40]. However, new records of *H. mohnikei* from Cambodia, Malaysia, China, Korea Thailand and Vietnam [37], along with recently published studies from southeastern India [11] and Singapore [41], have greatly expanded the known geographical range of *H. mohnikei* within Southeast Asia. As a consequence of suspected reduction in population size to the tune of >30% over the past 10 years due to fisheries

exploitation [37, 42–44], the status of *H. mohnikei* in the IUCN Red List of Threatened Species has been listed as 'Vulnerable' [45].

A probable specimen of *H. mohnikei* along the central west coast of India (CWCI) was recorded in February 2017 when a fisherman posted a picture of seahorse incidentally caught as bycatch in gill net operated in the Chapora estuary, Goa (India) (15.6120° N, 73.7506° E) on social media [S1 Fig] which was suspected to be an adult male of *H. mohnikei*. Sustained follow-up surveys for its occurrence in the surrounding environs of the Chapora estuary did not yield in any further specimens. Subsequently, four sub-adult specimens of *H. mohnikei*—hitherto sighted only from southeast coast of India [11], caught incidentally as bycatch in bag net fisheries during July, 2018 in the Mandovi estuarine system, Goa, CWCI have been examined here.

This contribution describes the first records of *H. mohnikei* from the western coastal India. To test further the hypothesis that the morphologically identified seahorse species is *H. mohnikei*, two mitochondrial DNA gene regions (loci), cytochrome oxidase subunit I (COI) and cytochrome *b* (Cyt *b*) were partially sequenced for reconfirmation. The potential role of ocean currents on the range expansion of *H. mohnikei* to north-western Asia has also been discussed. In the wake of the vulnerability of seahorse populations to threats such as habitat alteration/destruction and fishing pressure globally, the present sighting of *H. mohnikei* in a bay-estuarine system of CWCI is of considerable conservation and biogeographic significance.

## Materials and methods

### Collection site and seahorses

The locality (Brittona, Goa, India, 15.3059°N, 73.5073°E, Fig 1) at which the examined seahorse specimens were collected is a shallow water littoral environment in close proximity to the mangrove-dominated Chorao Island in the Mandovi estuary, CWCI. The currents are largely tide dominated, the tides being semi-diurnal with a mean amplitude of 1.3 m. Currents are also influenced by the large seasonal freshwater influx [46] during the summer monsoon season (June to September). Temporal variations in abiotic and biotic factors influenced by monsoonal precipitation and seasonal upwelling render the collection location as one of the most ecologically complex ecosystems [47,48]. Significant lowering of salinity occurs when the freshwater discharge is at its maximum during monsoon. During the post- and pre-monsoon periods, the salinity at the surface and bottom remains high (~25 ppt). The bottom topography at the collection site is muddy and comprises mainly silt and clay with rich organic matter content [49]. The collection site is reported to harbor rich and diverse fishery resources and its importance in supporting the life history stages of several marine teleosts has been well-documented [50].

The present report is based on four specimens of *H. mohnikei* (females, MK1 and MK2; males, MK3 and MK4; Figs 2 and 3) that were incidentally caught as bycatch in bag nets attached to fixed stakes (total length, 25 m; mouth length, 5 m; cod end mesh size, 10 mm) operated at a depth of ~6 m in the Mandovi estuary during July 2018. Dead seahorses landed as bycatch in bag nets attached to stakes operated by local fishermen in the estuarine system formed the study material. No live seahorses were specifically caught for the present study. Fin clips from these dead seahorses were used for molecular analysis. Since, the present study did not involve any experimentation with live specimens, the approval of the ethics committee for the usage of live animals for experiments is therefore inconsequential.

### Morphological analysis

Before subjecting them to a detailed analysis of morphological characteristics, seahorse specimens were carefully photographed (Nikon D7200, Japan) and sexed via observation of

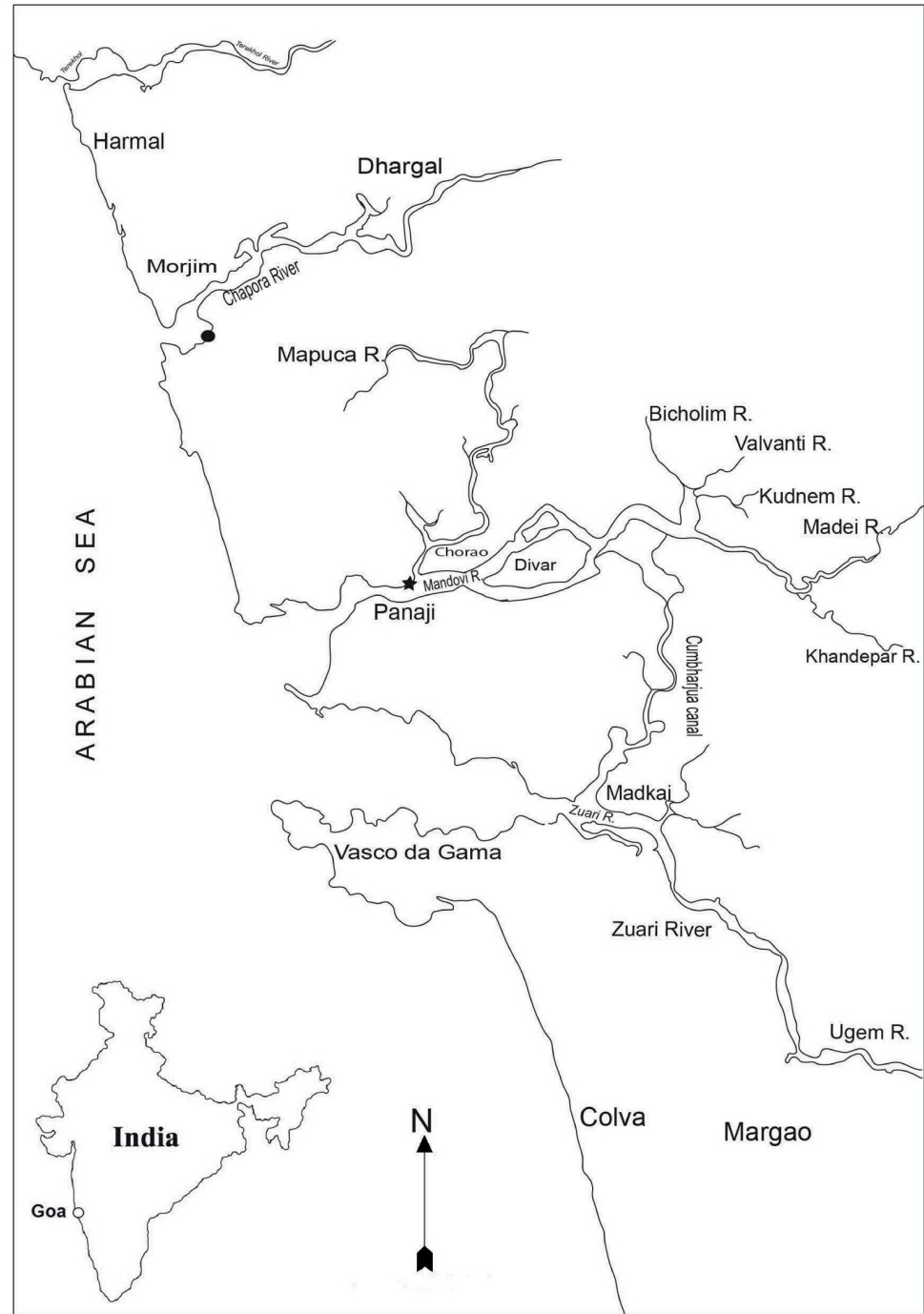

**Fig 1. Sampling localities of *H. mohnikei*.** A map showing major rivers and estuarine systems of Goa (central west coast of India). Location of a suspected adult male *H. mohnikei* in Chapora estuary (●); collection site of confirmed report of *H. mohnikei* in the Mandovi estuary (★).

presence or absence of brood pouch. Morphometric measurements and meristic counts were carried out following standardized protocols [15,17,51]. Morphometric characters such as height (Ht, linear distance from the top of the head to the tip of the stretched tail), head length (HL), trunk length (TrL), tail length (TaL) and standard length (SL = HL+TrL+TaL), were

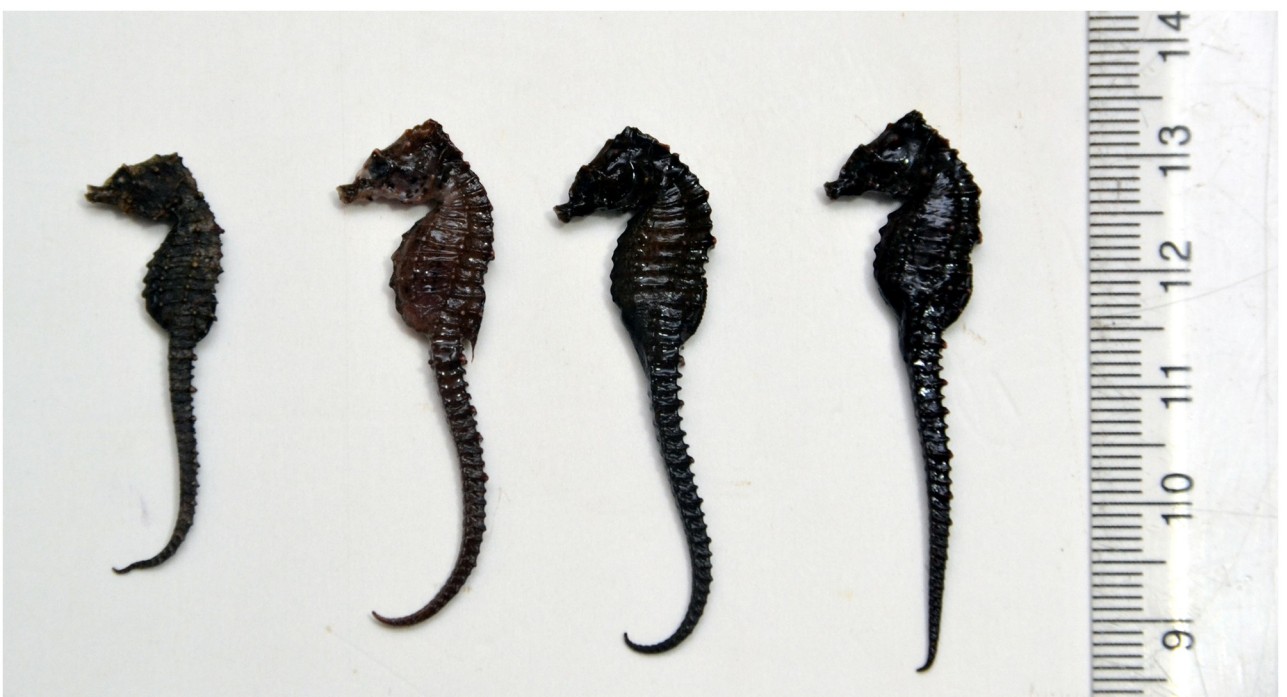

**Fig 2. Collected seahorses from Goa.** Sub-adult specimens of *Hippocampus mohnikei* (Bleeker 1853) (females, MK1 and MK2; males, MK3 and MK4; NIO1015/19) from the Mandovi estuary, Goa, central west coast of India.

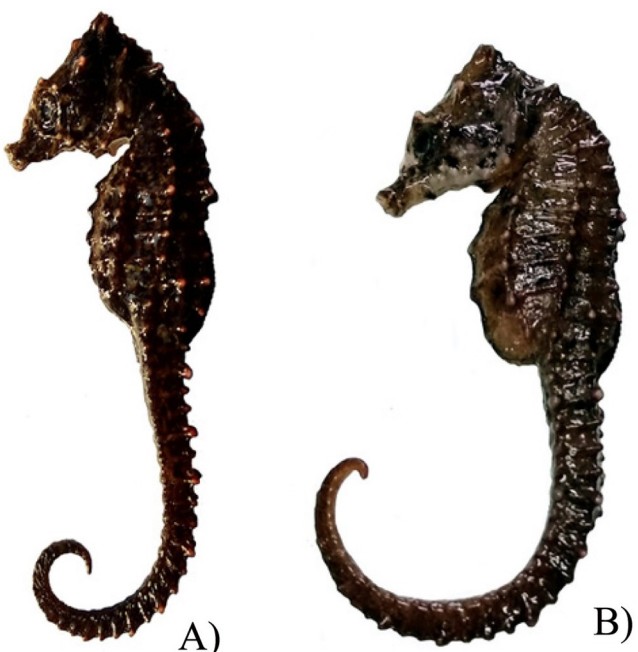

**Fig 3. Close-up view of *H. mohnikei*.** *Hippocampus mohnikei* (Bleeker 1853) (A-male (MK3) B-female (MK2) from the Mandovi estuarine system, Goa, central west coast of India (NIO1015/19).

measured. Measurements of snout length (SnL), ratio of head/snout lengths (HL/SnL), snout depth (SnD), head depth (HD), pectoral fin base length (PL) and dorsal fin base length (DL) were also conducted. All measurements were taken with the help of digital caliper to 0.1 mm and repeated to ensure accuracy. Trunk depths on 4[th] (TD4) and 9[th] (TD9) trunk rings were measured. The presence/absence of spines and dermal appendages was also noted. Meristic characters such as the number of trunk rings (TrR), tail rings (TaR), dorsal fin rays (DF), pectoral fin rays (PF) and anal fin rays (AF) were counted under stereo zoom microscope (Olympus SZX7, Japan) and the values were confirmed by triplicate counting. Brood pouch dimensions were measured in all male specimens.

Morphological identification of the collected seahorse specimens from Goa was based on comparison of the salient morphometric and meristic characters with the reported diagnostic features for different seahorse species [15,17,25]. For further confirmation, the morphological data of Goa specimens was compared with all vouchered specimens of *H. mohnikei* [11,15,25,52,53]. The voucher specimen collected from Goa is deposited in the national marine biodiversity repository at CSIR-National Institute of Oceanography, Goa, India (deposition ID: NIO1015/19).

## Statistical analysis

Morphological data of *H. mohnikei* and two other commonly occurring seahorse species (*H. kuda* and *H. trimaculatus*) in the coastal waters of India was evaluated by one-way analysis of variance (ANOVA) followed by the Duncan's multiple-range test [54]. The level of significance was tested at 5.0%, represented as $P < 0.05$. A comparison of morphological characters and meristic features of *H. mohnikei* (n = 4) with *H. kuda* (n = 10) and *H. trimaculatus* (n = 10) was made through principal component analysis (PCA). PCA was employed in which factor loadings based on components were used to determine the morphometric factors using PRIMER software (version 6) after square-root transformation.

## Molecular analysis: DNA extraction, PCR amplification and sequencing

For gene-based phylogenetic analysis, fragments of dorsal fin from freshly landed dead seahorses were dissected and preserved in 95% ethanol at –20°C until subjected to DNA extraction. DNA extraction was carried out as described by Kumar et al. with a few modifications [55]. In brief, lysis buffer and sodium dodecyl sulfate were added to the fin tissue and vortexed. Further, RNase and Proteinase K were added to the mixture and incubated for 15–20 minutes at room temperature. After the incubation period, the DNA was extracted using phenol:chloroform:isoamyl alcohol (25:24:1) mixture. Precipitated DNA was washed with 70% cold ethanol. The gel electrophoresis was performed to confirm the quality and integrity of DNA.

The seahorse mitochondrial cytochrome oxidase subunit I (COI) gene was amplified using primers, forward (5′ `TCAACTAATCACAAAGACATCGGCAC`3′) and reverse (5′ `ACTTCGGG GTGCCCAAAGAATC`3′) [56]. Similarly, the mitochondrial cytochrome *b* (Cyt *b*) gene fragment was also amplified using PCR with seahorse-specific primers, forward (5′ `AACYAGGACYAA TGRCTTGA`3′) and reverse (5′`GCASWAGGGAGGRKTTTAAC`3′) [43]. Reactions were carried out in a thermal cycler (Veriti® 96-well Thermal Cycler, Applied Biosystems, CA, USA). The amplification of COI and Cyt *b* genes was carried out by following cycling conditions as described by Lourie et al. with few modifications; an initial denaturation step of 95°C for 2 minutes followed by 39 cycles of 94°C for 30s, 55°C for 30s 72°C for 1 minute and a final extension at 72°C for 2 minutes [57]. Amplified PCR products of both COI and Cyt *b* genes were verified on agarose gel (1.5%) and purified for sequencing using PCR purification kit (Sigma Aldrich, USA). Gene sequence of the PCR products was determined using a taq dye deoxy terminator

cycle sequencing kit (PerkinElmer, CA, USA) following supplier's protocols. Sequencing reaction products were analyzed on 373 automated DNA Sequencer (Model 313 xl; Applied Biosystems, CA, USA). Amplification of the mitochondrial COI and Cyt *b* genes resulted in fragments of 609 bp and 639 bp, respectively.

## Phylogenetic analysis

Nucleotide sequences generated for four specimens collected from Goa (MK1, MK2, MK3 and MK4) were deposited in GenBank under accession numbers MN595216, MN595218, MN595217 and MK330041.1 for COI and MN595213, MN595214, MN595215 and MK112274.2 for Cyt *b* genes, respectively). Sequence integrity and genetic affinity with known species was compared using the basic local alignment search tool (BLAST) of NCBI [58] (http://www.ncbi.nih.gov/BLAST). To complete the analysis, sequences of Goa specimens were aligned with 47 and 85 sequences of COI and Cyt *b* genes, respectively of all vouchered specimens of *H. mohnikei* retrieved from the GenBank database (accession numbers shown in S1 Table). The mitochondrial COI and Cyt *b* gene sequences were aligned using the ClustalW algorithm implemented in MEGA 7 [59]. To ascertain stable and reliable relationships among the seahorse species, phylogenetic trees, separately for COI and Cyt *b* genes were constructed through maximum likelihood (ML) method using K2P distances. The analyses were performed in MEGA 7, and confidence levels were assessed using 1000 bootstrap replications. The alligator pipefish, *Syngnathoides biaculeatus* was used as an outgroup species. Genetic distance values are used to estimate intraspecies and interspecies kinships and also as a basis for phylogeny analysis. The degree of genetic divergence within *H. mohnikei* from different geographical locations (S1 Table) was assessed using Kimura 2-parameter (K2P) distance model [60] implemented in MEGA 7 [59]. Codon positions included were 1st+2nd+3rd+noncoding. All ambiguous positions with alignment gaps and missing data were removed from the sequences (pair-wise deletion option).

## Prevailing ocean circulation

The Indo-Pacific region (encompassing South China Sea, East China Sea, and southeastern Bay of Bengal including the Indian subcontinent) comes under the influence of strong seasonal monsoon wind reversal and associated reversals in the surface currents [61]. During the winter monsoon season (November to February), the surface circulation in the region indicates current pattern from east to west and the direction of the circulation reverses during the summer monsoon season (June to September). Considering the previous reported occurrences of *H. mohnikei* [37], the climatological currents in the region for the winter season which support a passive dispersal of marine fish species from east to west were considered for explaining the occurrence of *H. mohnikei* in Goa waters. Monthly mean climatology was prepared based on daily data on U (zonal) and V (meridional) components of geostrophic current velocities derived from Aviso merged product for the years, 2007–2017. Altimeter products by Ssalto/ Duacs and distributed by Aviso+, with support from Cnes (https://www.aviso.altimetry.fr) were used. The climatological currents generated for the winter season (November to February) from 2007 to 2017 for the Indo-Pacific region are depicted in Fig 4.

## Results

### Specimens examined

Colour of the head and body of freshly collected seahorse specimens (females, MK1 and MK2; males, MK3 and MK4; NIO1015/19) was dark brown to dark black. The entire body was

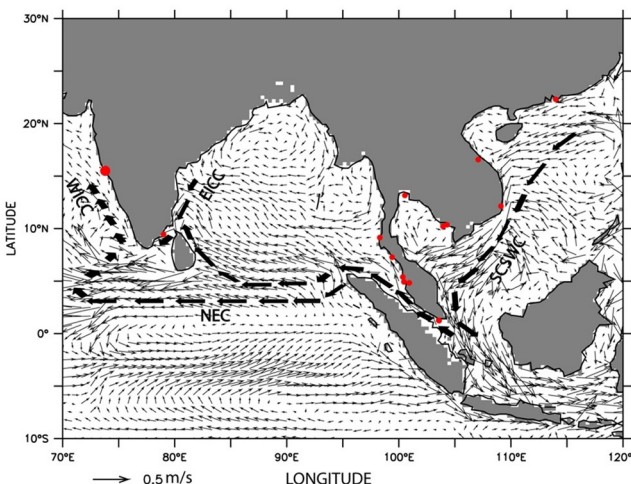

**Fig 4. Oceanic circulation in Indo-Pacific region.** Climatological surface currents during winter season (November–February) for the Indo-Pacific region. Red dots indicate the geographic locations of distribution of *Hippocampus mohnikei*. (WICC = West Indian Coastal Current, EICC = East India Coastal Current, NEC = North Equatorial Current and SCSWC = South China Sea Warm Current).

covered with low spines and the coronet was observed to be short with five tiny projections. Two prominent cheek spines and double rounded spines below the eye (each on the either side of the head) were discernible in all four specimens. (Figs 2 and 3).

## Morphological taxonomy

Measured height (Ht) and standard length (SL) of specimens varied between 39.2 and 54.2 mm (48.2 ± 6.5 mm) and 47.3 and 60.1 mm (54.95 ± 5.48 mm), respectively. TaL was observed to be relatively long compared to the TrL of the body. The ratio of the head length (HL) to snout length (SnL) of all four specimens varied between 3.2 and 3.5 (Table 1). Number of trunk rings (TrR) in all specimens were 11 whereas, the number of tail rings (TaR) varied narrowly between 37 and 39. The number of TrR supporting dorsal fin was two and the number of TaR supporting dorsal fin was counted to be one in all specimens. Number of dorsal fin pectoral fin and anal fin rays recorded were 15–16, 12–14 and 4, respectively. Slight enlargements, respectively at 1st, 4th, 7th and 10/11thTrR and 4th, 7/8th, 9/10th, 13/14thTaR were prominently discernible (Table 1).

## Comparative morphology

We compared the morphological features of *H. mohnikei* with other two commonly occurring seahorse species, *H. kuda* and *H. trimaculatus* occurring in coastal waters of India (Table 2). Fresh specimens of *H. kuda* was orange to dark brown with reddish spots over the body with single eye and cheek spine, rounded and backward curled coronet and deeper head and rounded or less developed body spines [15,16]. *H. trimaculatus* has three prominent black spots on the dorso-lateral surface of the first, fourth and seventh trunk rings [15,16] which are clearly absent in *H. mohnikei*. Single eye and hooked like cheek spine also observed in *H. trimaculatus*. Snout length (SnL) of *H. mohnikei* was shorter compared to the *H. kuda* and *H. trimaculatus*. These morphological characters clearly distinguish *H. mohnikei* from *H. kuda* and *H. trimaculatus* (Table 2).

**Table 1. Detailed morphology of *H. mohnikei*.** Morphological characteristics of four specimens of *Hippocampus mohnikei* collected from the Mandovi estuary, Goa, central west coast of India (MK1 and MK2 = Female; MK3 and MK4 = Male).

| | MK1 | MK2 | MK3 | MK4 |
|---|---|---|---|---|
| **Morphometric characters (mm)** | | | | |
| Height (Ht) | 39.2 | 54.2 | 48.1 | 51.3 |
| Standard length (SL) | 47.3 | 60.1 | 55.2 | 57.2 |
| Trunk length (TrL) | 16.1 | 19.4 | 15.2 | 15.1 |
| Tail length (TaL) | 24.2 | 32.1 | 31.2 | 33.1 |
| Head length (HL) | 8.5 | 9.1 | 9.2 | 8.7 |
| Snout length (SnL) | 2.5 | 3.1 | 3.3 | 2.5 |
| HL/SnL ratio | 3.4 | 3.2 | 3.3 | 3.5 |
| Snout depth (SnD) | 1.5 | 2.1 | 2.1 | 2.1 |
| Head depth (HD) | 5.5 | 7.1 | 6.9 | 7.2 |
| Eye orbit | 1.5 | 2.1 | 2.2 | 2.1 |
| Trunk depth between 4th and 5th trunk ring (TD4) | 3.1 | 4.2 | 3.9 | 4.1 |
| Trunk depth between 9th and 10th trunk ring (TD9) | 4.7 | 6.1 | 5.8 | 5.9 |
| Pectoral fin base length | 2.1 | 2.1 | 2.1 | 2.2 |
| Dorsal fin base length | 3.5 | 4.1 | 4.1 | 3.9 |
| Anal fin base length | 0.5 | 0.5 | 0.5 | 0.5 |
| Pouch length | - | - | 6.2 | 6.6 |
| Pouch depth | - | - | 3.1 | 2.5 |
| Pouch width | - | - | 3.2 | 3.3 |
| **Meristic features (numbers)** | | | | |
| Trunk Rings (TrR) | 11 | 11 | 11 | 11 |
| Tail Rings (TaR) | 37 | 38 | 37 | 39 |
| TrR supporting dorsal fin | 2 | 2 | 2 | 2 |
| TaR supporting dorsal fin | 1 | 1 | 1 | 1 |
| Cheek spines | 2 | 2 | 2 | 2 |
| Eye spines | 2 | 2 | 2 | 2 |
| Enlargement of trunk Rings | 1, 4, 7, 11 | 1, 7, 10 | 1, 4, 7, 11 | 1, 4, 7, 10 |
| Enlargement of tail Rings | 4, 7, 10, 13 | 4, 8, 10, 13 | 4, 7, 9, 13 | 4, 7, 10, 14 |
| Dorsal fin rays | 15 | 16 | 16 | 16 |
| Pectoral fin rays | 12 | 14 | 14 | 14 |
| Anal fin rays | 4 | 4 | 4 | 4 |

PCA used to elucidate the factors contributing to morphological differences in *H. mohnikei*, *H. kuda* and *H. trimaculatus* showed that many morphometric and meristic characters play an important role in differentiating these species. Results of PCA analysis revealed that 11 morphometric and meristic characters integrated into two principal components (PC1 and PC2) which cumulatively contributed to 97% of total variance (S2 Table). The PC1 which explained 81% of total variation consisted of morphometric characters such as SL, TaL, TD4, HL, TD9, TrL, HD, SnL and SnD while, the PC2 which explained 16% of total variation consisted of the TaR and DF. The derived ordination diagram after plugging standardized morphological data of all individuals of three seahorse species into PCA is depicted in Fig 5. PCA ordination diagram showed that three seahorse species were clustered out distinctly in orientation space. Therefore, based on comparison of morphological traits [15,16,25] and the results of PCA, the identity of seahorse specimens collected from Goa was confirmed as *H. mohnikei*.

**Table 2. Morphological comparison of seahorses.** Comparison of morphological characteristics of *H. mohnikei* with other two seahorse species, (*H. trimaculatus* and *H. kuda*). Sample size for each species shown in parentheses. Mean±SD denoted with different superscripts significantly differ from each other (P < 0.05).

| Morphometrics (mm) | *H. mohnikei* (n = 4) | *H. kuda* (n = 10) | *H. trimaculatus* (n = 10) |
|---|---|---|---|
| Height (Ht) | 48.2 ± 6.50[a] | 157.73 ± 19.24[b] | 93.06 ± 14.71[c] |
| Standard Length (SL) | 54.95 ± 5.48[a] | 182.65 ± 11.64[b] | 109.87 ± 6.45[c] |
| Trunk Length (TrL) | 16.45 ± 2.02[a] | 47.83 ± 2.54[b] | 26.40 ± 2.12[c] |
| Tail Length (TaL) | 30.15 ± 4.04[a] | 104.43 ± 9.07[b] | 61.70 ± 3.40[c] |
| Head Length (HL) | 8.83 ± 0.41[a] | 30.39 ± 1.88[b] | 21.70 ± 1.77[c] |
| Snout Length (SnL) | 2.67 ± 0.21[a] | 12.06 ± 1.04[b] | 8.90 ± 0.99[c] |
| HL/SnL ratio | 3.33 ± 0.14[a] | 2.54 ± 0.16[b] | 2.45 ± 0.19[bc] |
| Snout Depth | 1.95 ± 0.30[a] | 4.25 ± 0.41[b] | 3.45 ± 0.16[c] |
| Head Depth | 6.68 ± 0.79[a] | 14.80 ± 1.84[b] | 9.50 ± 0.53[c] |
| TD4 | 3.83 ± 0.50[a] | 12.16 ± 1.19[b] | 6.75± 0.86[c] |
| TD9 | 5.63 ± 0.63[a] | 18.68 ± 1.51[b] | 9.55 ± 0.90[c] |
| Pectoral fin base length | 2.03 ± 0.1[a] | 7.07 ± 1.2[b] | 4.75 ± 0.6[c] |
| Dorsal fin base length | 3.88 ± 0.3[a] | 15.73 ± 1.8[b] | 9.80 ± 0.8[c] |
| **Meristic counts** | | | |
| Trunk Rings (TrR) | 11 ± 0.00 | 11 ± 0.00 | 11 ± 0.00 |
| Tail Rings (TaR) | 38 ± 1.00[a] | 36 ± 1.00[b] | 41 ± 1.00[c] |
| TrR supporting Dorsal Fin | 2 ± 0.00 | 2 ± 0.00 | 2 ± 0.00 |
| TaR supporting Dorsal Fin | 1 ± 0.00 | 1 ± 0.00 | 1 ± 0.00 |
| Cheek Spines | 2 ± 0.00[a] | 1 ± 0.00[b] | 1 ± 0.00[c] |
| Eye spines | 2 ± 0.00[a] | 1 ± 0.00[b] | 1 ± 0.00[c] |
| Dorsal fin rays | 16 ± 1.00[a] | 18 ± 1.00[b] | 20 ± 1.00[c] |
| Pectoral fin rays | 14 ± 1.00[a] | 16 ± 1.00[b] | 17 ± 1.00[c] |

Morphological measurements revealed that all the four Goa specimens are consistent and congruent with the salient morphological descriptions for type specimens of *H. mohnikei* reported by Lourie et al. [15,17,25] for specimens collected from Vietnam and Japan. Furthermore, meristic, morphometric, and key diagnostic morphological character comparisons from vouchered specimens of *H. mohnikei* [11,15,25,52,53], reconfirmed the identity of seahorse specimens collected from Goa as *H. mohnikei* (Table 3).

## Phylogenetic analysis and genetic distance

PCR for the mitochondrial COI and Cyt *b* genes of Goa seahorse specimens resulted in 609 bp and 639 bp amplicons, respectively. Maximum likelihood (ML) phylogenetic analysis were performed using K2P distances with sequences of COI and Cyt *b* genes of seahorse specimens from Goa (MK1, MK2, MK3 and MK4) with all downloaded sequences of vouchered *H. mohnikei* from GenBank (S2 Table). ML trees of Goa seahorse specimens (GenBank accession numbers, MK330041.1, MN595216.1, MN595217.1 and MN595218.1 for COI gene; GenBank accession numbers, MK112274.2, MN595213.1, MN595214.1 and MN595215.1 for Cyt *b* gene) showed that all four individuals clustered together suggesting a monophyletic lineage (Figs 6 and 7). Furthermore, sequences of COI and Cyt *b* genes of Goa specimens, respectively showed maximum phylogenetic similarities with *H. mohnikei* (GenBank accession numbers GQ502157.1) (GenBank accession numbers: EU179923.1 and EU179924.1) strongly supported by high bootstrap values of >95% (Figs 6 and 7).

Genetic distance describes the kinship distance within and between species. Calculated pair-wise genetic distance (K2P) values based on COI and Cyt *b* sequences of *H. mohnikei*

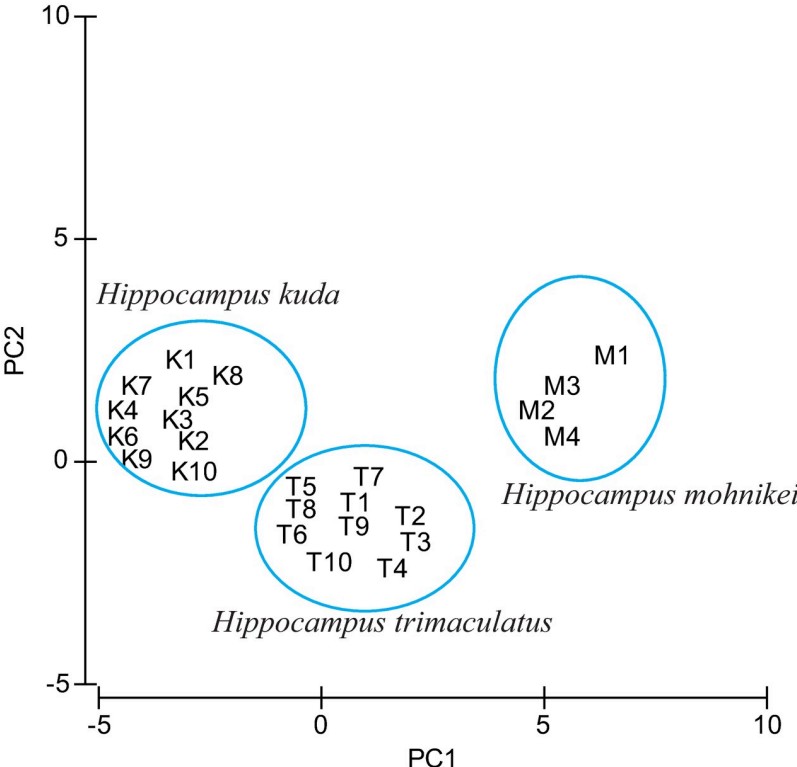

**Fig 5. PCA analysis scores.** Scores of factor loadings plot of principal component analysis of three species, *H. mohnikei*, *H. kuda* and *H. trimaculatus* based on morphological data.

from Goa and other localities (Japan, Korea, China, Vietnam and Thailand) revealed that *H. mohnikei* from Goa the show least genetic distances of 0.7% for both COI and Cyt *b* genes with populations of *H. mohnikei* from Vietnam (COI) and Thailand (Cyt *b*), respectively (Tables 4 and 5). In contrast, Goa specimens showed highest genetic distances of 1% for COI and 7% for Cyt *b* genes with populations of *H. mohnikei* from China and Japan, respectively.

## Ocean circulation

As seen from Fig 4 that the circulation in the Indo-Pacific region shows a large-scale flow moving from the East China Sea to the South China Sea. From the South China Sea, the major part of the flow turns southeastward entering into the Java Sea and a small part turns northwestward to eastern Bay of Bengal (BoB) and the Andaman Sea through Malacca Strait. From eastern BoB, the surface waters flow westward as a part of the north equatorial current (NEC). On encountering the coast of Sri Lanka, a part of the NEC move northward into the northern BoB, while the rest moves westward and into the southern Arabian Sea, south of Sri Lanka. The NEC feeds into the wintertime west India coastal current (WICC) and moves waters from NEC towards north into the Arabian Sea along its eastern boundary.

## Discussion

The geographic range of *Hippocampus mohnikei* is still unresolved as Lourie et al [15,16] confirmed the nativity of this species to Japan and Vietnam and likely extending throughout the South China Sea and the Gulf of Thailand [16,17,25]. New records on the occurrence of *H.*

**Table 3. Comparison of morphological characters.** Comparison of diagnostic morphological characters (range and mean) of vouchered specimens of *Hippocampus mohnikei*. Values in parentheses are mean.

| Data source | Present Study | Lourie et al. [15,25] | Thangaraj and Lipton [11] | Zhang et al. [52] | Wibowo et al. [53] |
|---|---|---|---|---|---|
| Voucher/field Number | NIO1015/19 | RMNH7259 | CMFRI No. 146 | SCSMBC007418–19 | KAUM-1. 17724 |
| Locality | Goa, west coast of India | Vietnam | Southeast coast of India | Yellow Sea, China | Kagoshima, Southern Japan |
| No. of samples (N) | 04 | 04 | 01 | 16 | 01 |
| **Morphometrics (mm)** | | | | | |
| Standard Length (SL) | 47.3–60.1 (54.95) | 54.0–57.0 | 70.0 | 52.5–83.0 (66.4) | - |
| Trunk Length (TrL) | 15.1–19.4 (16.45) | | 22.0 | 15.0–23.0 (19.2) | - |
| Tail Length (TaL) | 24.2–33.1 (30.15) | | 38.0 | 28.5–46.0 (36.7) | - |
| Head Length (HL) | 8.5–9.2 (8.83) | | 10.0 | 9.0–14.0 (10.5) | - |
| Snout Length (SnL) | 2.5–3.3 (2.67) | | 3.0 | 1.8–3.9 (2.4) | |
| HL/SnL ratio | 3.2–3.5 (3.35) | 2.8–3.9 (3.0) | 3.33 | 3.59–5.75 (4.5) | 3.1 |
| Snout Depth | 1.5–2.1 (1.95) | - | 2.5 | - | - |
| Head Depth | 5.5–7.2 (6.68) | - | 7.0 | - | - |
| TD4 | 3.1–4.2 (3.83) | - | 5.5 | - | - |
| TD9 | 4.7–6.1 (5.63) | - | 8.0 | - | - |
| **Meristic counts** | | | | | |
| Trunk Rings (TrR) | 11 | 11 | 11 | 11 | 11 |
| Tail Rings (TaR) | 3737–39 (38) | 37–40 (38) | 38 | 37–40 (38) | 38 |
| TrR supporting dorsal fin | 2 | 2 | - | - | - |
| TaR supporting dorsal fin | 1 | 1 | - | - | - |
| Cheek Spines | 2 | 2 | 2 | | 2 |
| Rounded spines below eye | Double | Double | Single | - | - |
| Dorsal fin rays (DF) | 15–16 | 15–16 | 14 | 16 | 15 |
| Pectoral fin rays | 12–14 | 12–14 (13) | 12 | 12–14 (13) | 12 |
| Anal fin rays (AF) | 4 | - | - | 4 | - |

*mohnikei* from many other locations beyond on its native range in north- and south-eastern coastal waters of Asia [11,37,41,44,56,62] suggest its significant extended distribution in the Indo-Pacific region (Fig 8). Records of *H. mohnikei* in the Palk Bay, southeast coast of India [11] is to date the known range limit towards the west. In southeastern Asian coastal waters, *H. mohnikei* has been reported from Singapore [41] as well as from Cambodia, Malaysia, Thailand and Vietnam [37]. In northeast Asia, *H. mohnikei* is reported from China [44,56] and South Korea [62,63].

The present report on the occurrence of *H. mohnikei* in a bay-estuarine system of Goa, CWCI as confirmed by both morphological and molecular analysis represents a significant westward expansion compared to the previously known geographic range of this species [37]. Since the reporting of *H. mohnikei* solely based on morphological characters from the Palk Bay, southeast coast of India [11], no further sightings of this species have been reported from coastal marine waters of India. Therefore, a comparison of genetically distinct populations of *H. mohnikei* inhabiting native (Japan and Vietnam) and extended ranges in the Indo-Pacific region (Fig 8) with Goa specimens gains significance. Lourie et al. reported that a Vietnamese specimen of *H. mohnikei* was genetically distinct from *H. mohnikei* from Japan [16,25].

A maximum likelihood (ML) tree constructed of Goa specimens with all vouchered specimens of *H. mohnikei* based on COI gene using K2P distances showed maximum phylogenetic proximity with a specimen from Vietnam (GenBank accession number, GQ502157.1) (Fig 6). On the other hand, ML tree constructed with sequences of Cyt *b* gene of Goa specimens showed maximum genetic similarities with *H. mohnikei* from Thailand (GenBank accession

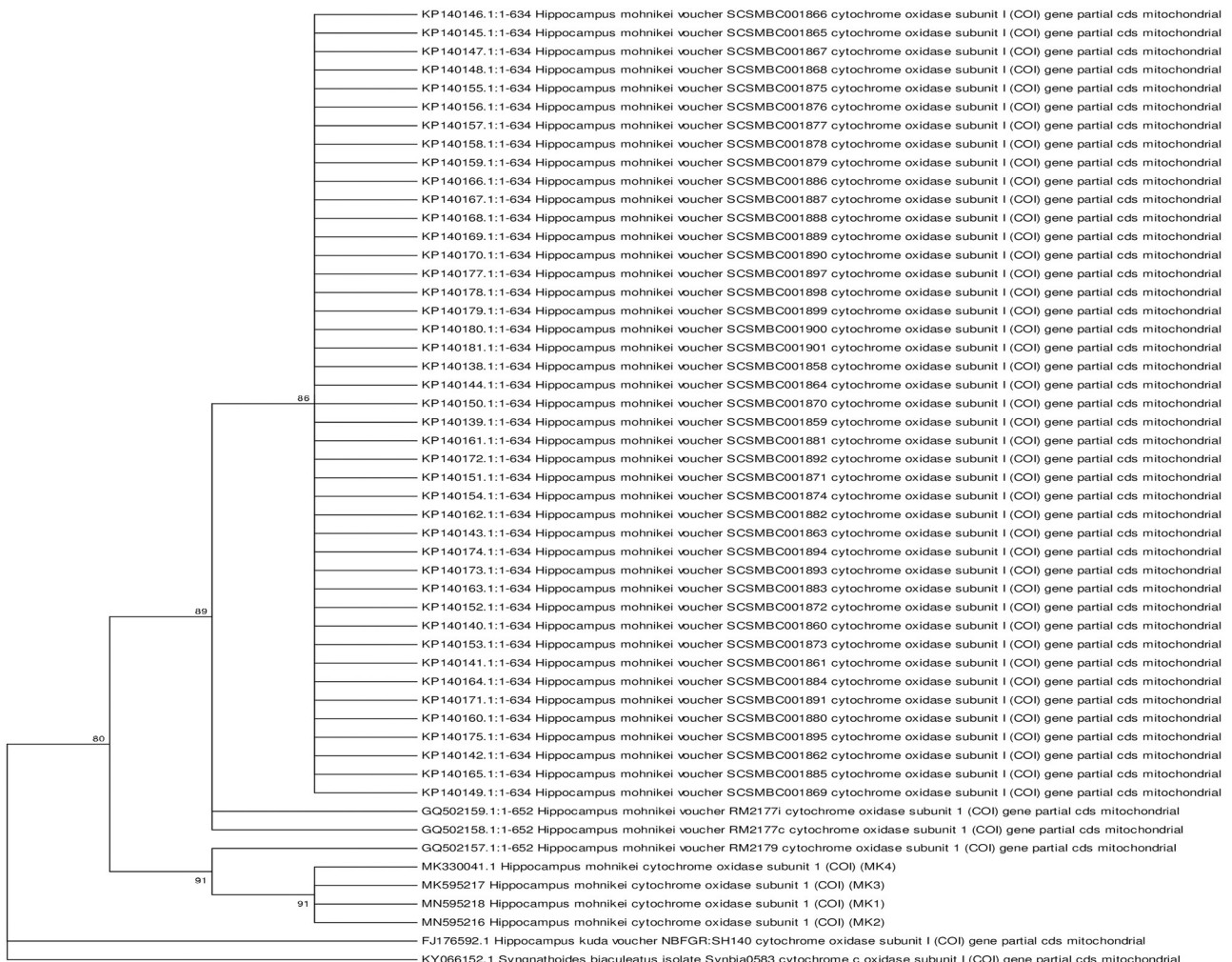

**Fig 6. Phylogenetic relationships of Goa specimens with other seahorse species based on COI gene sequences.** ML phylogenetic tree of Goa specimens and all vouchered *Hippocampus mohnikei* based on mitochondrial cytochrome oxidase subunit I gene (COI) sequences using K2P distances with 1000 bootstrap iterations. Numbers at nodes indicate bootstrap support values and nodes with >50% bootstrap support values are shown. The alligator pipefish, *Syngnathoides biaculeatus* (KY066152.1) served as an out-group species. Scale bar = genetic distance of 0.05. ML = Maximum likelihood.

numbers, EU179923.1 and EU179924.1) (Fig 7). In both ML trees of COI and Cyt *b* genes, phylogenetic proximity of all four Goa individuals with >95% bootstrap support thus confirming the taxonomic identity of Goa specimens as *H. mohnikei*. Furthermore, a comparison of pair-wise genetic distance values (K2P) based on sequence COI (Table 4) and Cyt *b* (Table 5) genes showed that Goa specimens showed least genetic distance with populations of *H. mohnikei* from Vietnam (K2P value, 0.7%) and Thailand (K2P value, 0.7%), respectively. In contrast, Goa specimens showed highest genetic distances of 1% for COI and 7% for Cyt *b* genes with populations of *H. mohnikei* from China and Japan, respectively. Similar high pair-wise K2P values of sequences of Cyt *b* between *H. mohnikei* populations from the Gulf of Thailand and those from China (13.54–14.46%) and Japan (14.42–14.79%) have been recently reported [64].

A comparative assessment of K2P values of the present study with those from Vietnam [65] and Gulf of Thailand [64], respectively for COI and Cyt *b* suggest that *H. mohnikei* populations from Goa, Vietnam and the Gulf of Thailand might share a similar genetic identity and

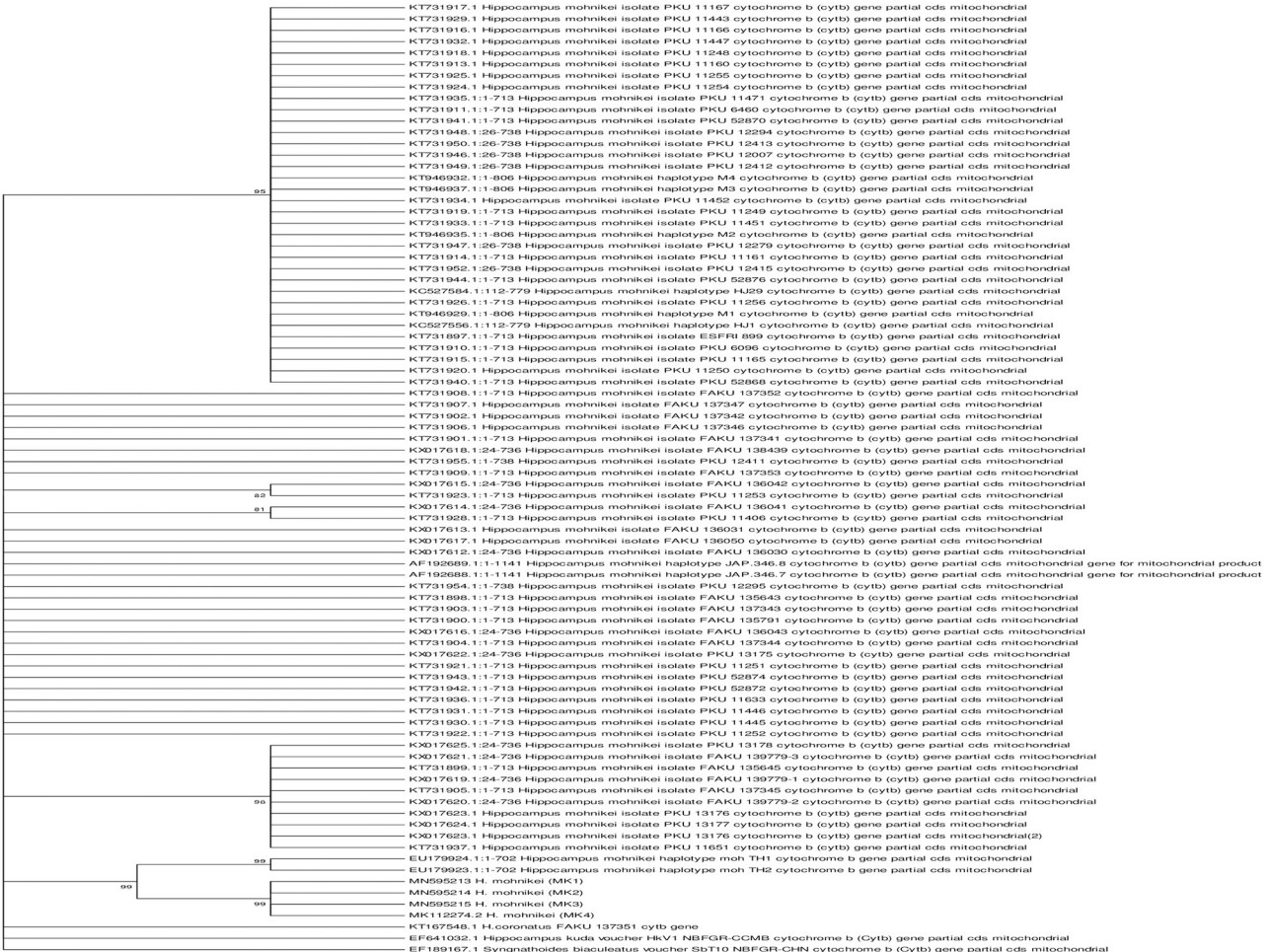

**Fig 7. Phylogenetic relationships of Goa specimens with other seahorse species based on Cyt *b* gene sequences.** ML phylogenetic tree of Goa specimens and all vouchered *Hippocampus mohnikei* based on mitochondrial cytochrome *b* gene (Cyt *b*) sequences using K2P distances with 1000 bootstrap iterations. Numbers at nodes indicate bootstrap support values and nodes with >50% bootstrap support values are shown. The alligator pipefish, *Syngnathoides biaculeatus* (EF189167.1) served as an out-group species. Scale bar = genetic distance of 0.05. ML = Maximum likelihood.

genetically distinct from those from China and Japan. Also, a possible genetic differentiation in populations of *H. mohnikei* inhabiting coastal marine waters of northeastern Asia (Japan, Korea and China) and southeastern Asia (Vietnam, Thailand, Cambodia, Malaysia, Singapore and India) may not be ruled out. Furthermore, least genetic divergence recorded between different populations of *H. mohnikei* (Goa vs Vietnam, COI gene) and (Goa vs. Thailand, Cyt *b* gene), the possibility of crypticity within *H. mohnikei* populations from these localities is ruled

**Table 4. Pair-wise genetic distance based on COI gene sequences.** Pair-wise genetic distances (Kimura 2-parameter model) between *Hippocampus mohnikei* populations of Goa and the Indo-Pacific region based on COI gene sequences. Values in parentheses indicate percentage.

| Localities of *H. mohnikei* with GenBank accession numbers | Goa | Japan | Vietnam | China |
|---|---|---|---|---|
| Goa Specimens (MN595216–MN595218; MK330041.1) | 0.00 | | | |
| Japan (GQ502158.1–GQ502159) | 0.008 (0.8) | 0.00 | | |
| Vietnam (GQ502157.1) | 0.007 (0.7) | 0.009 (0.9) | 0.00 | |
| China (KP140138–KP140181) | 0.010 (1.0) | 0.003 (0.3) | 0.010 (1.0) | 0.00 |

**Table 5. Pair-wise genetic distance based on Cyt *b* gene sequences.** Pair-wise genetic distances (Kimura 2-parameter model) between *Hippocampus mohnikei* populations of Goa and the Indo-Pacific region based on Cyt *b* gene sequences. Genetic divergence of > 2% within same locality separated out. Values in parentheses indicate percentage.

| Localities of *H. mohnikei* with GenBank Accession numbers | Goa | Japan1 | Japan2 | Korea1 | Korea2 | China | Thailand |
|---|---|---|---|---|---|---|---|
| Goa Specimens (MN595213.1, MN595214.1, MN595215.1 and MK112274.2) | 0.00 | | | | | | |
| Japan1 (AF192688.1–;AF192689.1; KT731898.1–;KT731909.1) | 0.067 (6.7) | 0.00 | | | | | |
| Japan2 (KX017612.1–;KX017625.1) | 0.070 (7.0) | 0.071 (7.1) | 0.00 | | | | |
| Korea1 (KT731897.1–;KT731920.1; KT731924.1–;KT731926.1; KT731929.1; KT731932.1–; KT731935.1; KT731940.1–;KT731941.1; KT731944.1–KT731952.1 and KT731955.1) | 0.066 (6.6) | 0.040 (4.0) | 0.044 (4.4) | 0.00 | | | |
| Korea2 (KT731921.1–KT731923.1; KT731928.1; KT731930.1–KT731931.1; KT731936.1– KT731937.1, KT731943.1 and KT731955.1) | 0.067 (6.7) | 0.004 (0.4) | 0.012 (1.2) | 0.066 (6.6) | 0.00 | | |
| China (KT946929.1–;KT946937.1; KC527556.1 and KC527584.1) | 0.066 (6.6) | 0.037 (3.7) | 0.042 (4.2) | 0.057 (5.7) | 0.004 (0.4) | 0.00 | |
| Thailand (EU179923.1–;EU179924.1) | 0.007 (0.7) | 0.001 (0.1) | 0.001 (0.1) | 0.001 (0.1) | 0.001 (0.1) | 0.001 (0.1) | 0.00 |

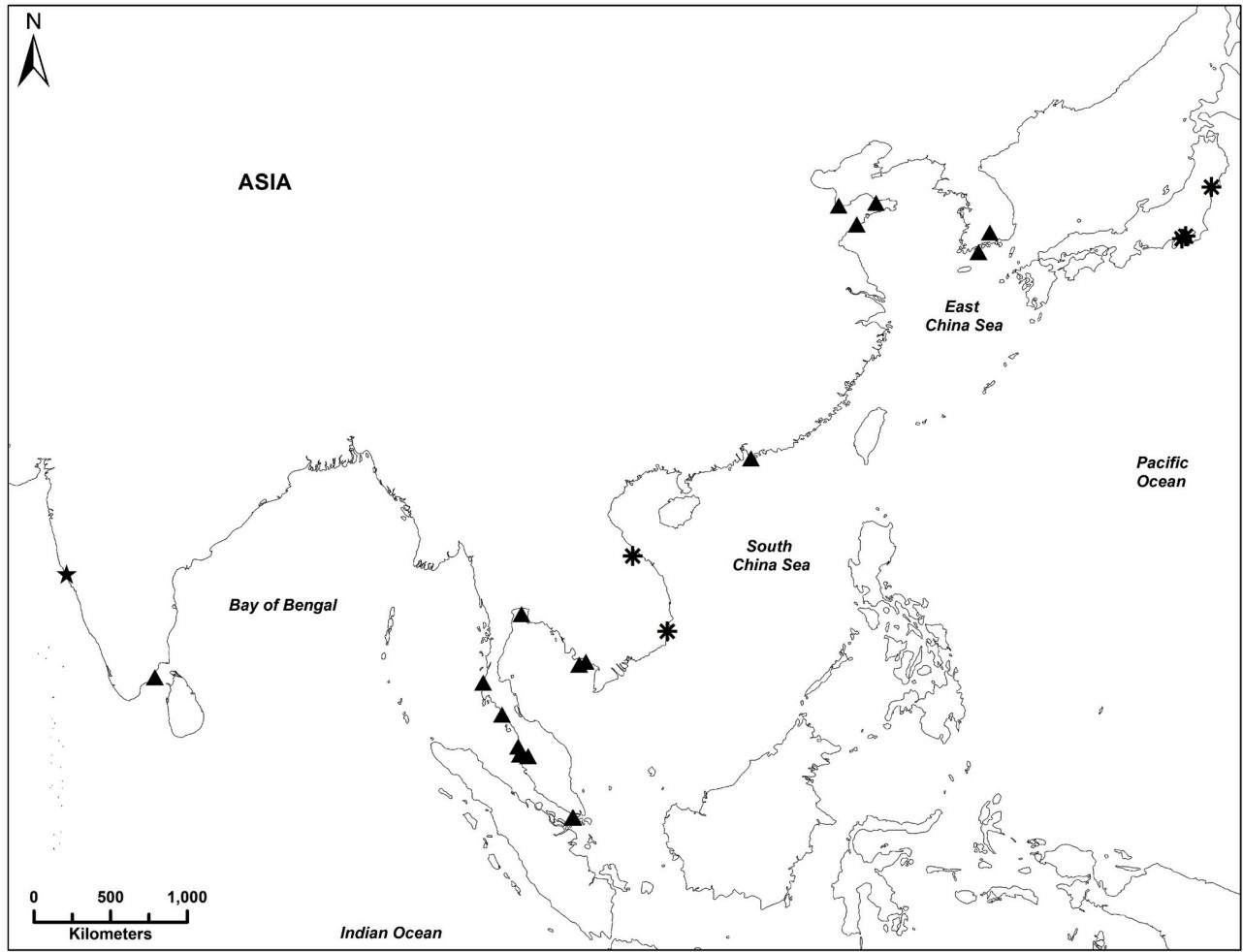

**Fig 8. Reported sightings of *H. mohnikei* in Indo-Pacific region.** Map showing native range, recent sightings and all reported sightings for *Hippocampus mohnikei* in the Indo-Pacific region. Sightings are classified as new (this study, ★), recently published (2004–2016, ▲) and original records [15] (*). Map was modified from Aylesworth et al. [37] and re-created in the World Geodetic System (WGS) 84 datum.

out by applying 2% threshold genetic divergence for assessing intra-specific differences [16]. In contrast, recent study by Panithanarak et al observed high genetic variance between the populations of *H. mohnikei* from the Gulf of Thailand and China (13.54–14.46%) and Japan (14.42–14.79%) suggested the possibility of cryptic species within the known range of *H. mohnikei* [64]. Further studies, however, are necessary to assess the genetic variation within *H. mohnikei* populations in the entire Indo-Pacific region.

Recent studies have advanced our understanding of habitat associations of *H. mohnikei* compared to what has been described in the IUCN Red List Assessment [45]. Previously, *H. mohnikei* was reported only in association with seagrass beds in Japan and in an estuary in Vietnam [16,25]. Association of *H. mohnikei* populations with seagrass beds (*Zostera* spp.) in Korea [62,63], Japan [66] and China [44] has been reported. Whereas the specimen reported from Palk Bay, southeastern India [11] was associated with two other additional genera of seagrass species (*Halophila* and *Thalassia*). Recently, published data on the habitat preferences of *H. mohnikei* from Southeast Asia [37] indicate its association with seagrass beds of *Cymodocea* spp., *Enhalus* spp., *Halodule* spp., *Halophila* spp., *Syringodium* spp. and *Thalassia* spp. Furthermore, these authors also report the association of *H. mohnikei* with mangroves, oyster beds, mussel farms and sandy beaches, whereas, Lim reported its occurrence in association with seaweeds in Singapore [41]. From these studies, it is evident that *H. mohnikei* occupies a wide variety of habitats.

All previous reports on the habitat descriptions of *H. mohnikei* were located in the coastal waters and bay-estuarine systems in the depth range of 0–10 m [11,17,25,37,44,62]. Interestingly, the occurrence of small juveniles of *H. mohnikei* in association with highly dense live prey organisms in the offshore waters of Japan has also been reported [39]. The present report of *H. mohnikei* is based on specimens caught as an incidental catch in bag nets fixed to the stakes operated in the Mandovi estuary, Goa. Although precise information on the habitat of *H. mohnikei*, could not be ascertained, it is worth describing the surrounding environs at the site of collection. Predominantly, the estuarine stretches of the River Mandovi have dense mangrove coverage and extensive oyster beds [67] and patches of seagrass, *Halophila ovalis* and *H. beccarii* [68]. The channel network of Mandovi estuary is characterized by islands (Chorao, Divar and Vanxim) and many sheltered areas interspersed with dense mangrove ecosystem. The present occurrence of *H. mohnikei* in a bay-estuarine system of Goa is not quite surprising as these are the typical habitats where this species has been previously reported. For ascertaining habitat shift by geographically disconnected populations of *H. mohnikei*, further intense studies on habitat specificities of different life stages therefore, are warranted.

Food availability has been considered as one of the factors influencing the abundance and distribution of seahorses [20]. Zooplankton such as *Oithona davisae*, Calanoida and *Penilia avirostris* have been identified as the main prey organisms for juvenile and young *H. mohnikei* in Tokyo Bay [39], whereas the adults have been reported to prey upon mainly larger organisms such as gammarid and caprellid amphipods [63]. Nutrient-rich waters of the Mandovi estuarine system sustain high primary productivity and abundance of zooplanktonic prey organisms such as copepods [69–71] which are essential for the survival and establishment of seahorse populations. Considering the prevailing environmental conditions, mangrove shelter and abundance of prey organisms, the likelihood of establishment and colonization of seahorse populations in these sheltered areas appears high.

Factors such as habitat, diet, and even anthropogenic activities are the main reasons cited for the migration of seahorses [72]. Furthermore, instances of seahorses for long distance dispersal by means of rafting along with seaweeds or any other suitable holdfast under the influence of ocean currents during juvenile stages have been documented [2,22]. A southward dispersal of *H. histrix* ~1800 km under the influence of East Australian Current in the Great

Barrier Reef has been reported [33]. Furthermore, ~1100 km dispersal from the south to the northeast coast of India by *H. fuscus* under the influence of East India Coastal Current has been also reported [35]. A Direct evidence for passive long-distance dispersal of *H. trimaculatus* at Malacca Strait has been documented recently [36].

Newborn juveniles of many seahorse species undergo a pelagic phase (planktonic) ranging between two and six weeks depending on the species before they settle down into sessile habitats (benthos) [2,73]. Consequently, planktonic phase juveniles are likely to facilitate in widespread gene flow across geographically disconnected populations and resulting in genetic homogeneity [74]. The occurrence of small juveniles of *H. mohnikei* (15.1–45.5 mm) in association with highly dense live prey organisms in offshore waters of Japan [39] suggests that food availability is essential for sustaining survivorship of planktonic phase juveniles. Furthermore, a seasonal (warm and cold season) inshore-offshore migration by *H. mohnikei* populations inhabiting Chinese coast has been confirmed [56]. Higher passive dispersal capabilities of seahorse juveniles with floating debris, prey swirls and with seaweed rafts than adults has been reported [2,30,36]. It has been postulated that marine species with greater longevity, free swimming and feeding (planktotrophic) larval phases have relatively high dispersal abilities and thus permit genetic exchange between populations [75]. It has been hypothesized that owing to lack of suitable settlement habitat (holdfasts) and or scarcity of larger food items may render larger non-settled juveniles of *H. mohnikei* to prolong their planktonic phase [39]. Therefore, an extension planktonic phase as observed in case of *H. mohnikei* juveniles in offshore waters of Tokyo Bay [39] may entail higher degree of passive dispersal under the influence of oceanic circulation, especially when they are attached to floating debris and algae.

The confirmed occurrence of four sub-adult specimens (Mandovi estuary) and suspected occurrence of an adult specimen of *H. mohnikei* in the proximately adjacent bay-estuarine ecosystem (Chapora estuary) of Goa, CWCI could be a consequence of either long-distance dispersal from Southeast Asian countries or stepping-stone mediated dispersal as revealed by oceanic currents circulation in the Indo-Pacific region. It is clearly seen from Fig 4 that water mass from southeastern Asian coasts and east coast of India enters into the Arabian Sea as a result of winter circulation [61]. In the absence of geographical barriers between *H. mohnikei* populations of Goa and Southeast Asian countries, ocean currents flowing from regions where sightings of *H. mohnikei* may contribute to wide-spread dispersal. Therefore, it is plausible that oceanic currents may be aiding long-distance or stepping stone dispersal of *H. mohnikei* populations from different geographical locations such as Cambodia, Malaysia, Vietnam and Thailand [15,16,37,64], Singapore [41] and the Palk Bay, southeast coast of India [11] to the west coast of India.

Conversation with local fishermen who operate stake nets at the collection site revealed that sightings of seahorses are relatively more common during summer monsoon season than in non-monsoonal months. During the summer monsoon period, a number of rivulets and rivers connected to the Mandovi estuarine network discharge large amounts of freshwater [76]. Consequently, it is possible that the rainwater discharge from sheltered areas—the preferred seahorse habitats, may force seahorses flow downstream along with currents due to their poor swimming abilities.

It is well known that the west coast of India comes under the influence of substantial coastal cold-water upwelling event during the summer monsoon season [47,76]. It has been reported that the survival and dispersal of tropical marine species are known to be impacted by a drastic decrease in water temperature associated with upwelling [77–79]. Consequently, seahorse populations in western coastal India might undergo seasonal fluctuations [22]. A significant decrease in abundance and diversity of demersal fishes during the summer monsoon in a bay-estuarine system of the Mandovi river has been documented [50]. As a consequence of the

reduction in salinity and drop in temperature associated with monsoonal precipitation and upwelling, respectively the seahorse populations might face limitations in growth and recruitment along the west coast of India. Interestingly, unlike most subtropical seahorse species, the greater tolerance by native populations of *H. mohnikei* to cold water in the Western Pacific has been reported [31]. However, further temperature stress simulation studies are required to ascertain whether geographically disconnected populations of *H. mohnikei* also possess a similar trait or not.

## Conclusions

Based on the new information presented in this paper, there is clear evidence that *H. mohniei* has expanded its natural and reported geographic range. Recognizing the potential of seahorses as a popular flagship species in estuarine conservation and sustainable management [1,80,81], significant westward range expansion of *H. mohnikei* is of considerable biogeographic and conservation importance. Intriguingly, the possibility of *H. mohnikei* originating in India and subsequent dispersal to the east cannot be ruled out. Although there is no published evidence to show that *H. mohnikei* originated in Japan/China, the original type specimens studied were however, from Japan and Vietnam. At the same time, it may be plausible that *H. mohnikei* might be present in other localities, but has not been reported. Further intense sampling and rigorous molecular analysis of *H. mohnikei* populations along the Indian coast and throughout its range would be therefore required to confirm whether these are vagrants or a previously unreported, established population.

## Supporting information

**S1 Fig. Picture of *H. mohnikei* posted on social media.** Male specimen of *Hippocampus mohnikei* caught by gill net in the Chapora estuary, Goa, India.
(TIF)

**S1 Table. The COI and Cyt *b* gene sequences.** GenBank accession numbers and sources of the mitochondrial gene sequences (COI and Cyt *b*) of seahorse species used for construction of phylogenetic trees.
(DOC)

**S2 Table. The COI and Cyt *b* gene sequences.** GenBank accession numbers and sources of the mitochondrial gene sequences (COI and Cyt *b*) of seahorse species used for construction of phylogenetic trees.
(DOCX)

## Acknowledgments

The authors are grateful to Prof. Sunil Kumar Singh, Director, CSIR-National Institute of Oceanography, Goa (India) for encouragement and facilities. Thanks are due to local fisherman Mr. Harish Halarnkar for providing seahorse specimens. Thanks are also due to Dr. Mani Murali and Mr. Devanand Kavlekar for their help in preparing the maps. We would like to deeply thank Dr. Qiang Lin for sharing the morphometric data of *Hippocampus mohnikei* for comparison. The authors are thankful to Riley Pollom and Dr. Graham Short and two anonymous reviewers for their critical and constructive comments for improving the quality and presentation of the manuscript. This represents contribution No. 6515 of the CSIR-National Institute of Oceanography, Goa (India).

## Author Contributions

**Conceptualization:** Sushant V. Sanaye, Anantha Sreepada Rayadurga.

**Data curation:** Rakhee Khandeparker, Harshada Kankonkar.

**Formal analysis:** Sushant V. Sanaye, Rakhee Khandeparker, Anantha Sreepada Rayadurga, Mamatha S. Shivaramu, Jayu Narvekar, Mukund Gauthankar.

**Funding acquisition:** Anantha Sreepada Rayadurga.

**Investigation:** Sushant V. Sanaye, Anantha Sreepada Rayadurga, Harshada Kankonkar, Mukund Gauthankar.

**Methodology:** Sushant V. Sanaye, Rakhee Khandeparker, Mamatha S. Shivaramu, Harshada Kankonkar, Jayu Narvekar, Mukund Gauthankar.

**Project administration:** Rakhee Khandeparker, Anantha Sreepada Rayadurga.

**Resources:** Rakhee Khandeparker, Mamatha S. Shivaramu, Jayu Narvekar.

**Supervision:** Rakhee Khandeparker, Anantha Sreepada Rayadurga, Mamatha S. Shivaramu.

**Validation:** Mamatha S. Shivaramu, Jayu Narvekar.

**Visualization:** Anantha Sreepada Rayadurga.

**Writing – original draft:** Sushant V. Sanaye.

**Writing – review & editing:** Anantha Sreepada Rayadurga.

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
