## [Decision Letter · Decision Letter 0]

24 Sep 2019

PONE-D-19-19415

Morphological and molecular evidence for range extension and first occurrence of the Japanese seahorse, Hippocampus mohnikei (Bleeker 1853) in a bay-estuarine system of Goa, central west coast of India

PLOS ONE

Dear Mr Rayadurga,

Thank you for submitting your manuscript to PLOS ONE. After careful consideration, we feel that it has merit but does not fully meet PLOS ONE’s publication criteria as it currently stands. Therefore, we invite you to submit a revised version of the manuscript that addresses the points raised during the review process.

We would appreciate receiving your revised manuscript by Nov 08 2019 11:59PM. To enhance the reproducibility of your results, we recommend that if applicable you deposit your laboratory protocols in protocols.io, where a protocol can be assigned its own identifier (DOI) such that it can be cited independently in the future. For instructions see: http://journals.plos.org/plosone/s/submission-guidelines#loc-laboratory-protocols

We look forward to receiving your revised manuscript.

Kind regards,

Rui Rosa

Academic Editor

PLOS ONE

Journal Requirements:

The name of the colleague or the details of the professional service that edited your manuscriptA copy of your manuscript showing your changes by either highlighting them or using track changes (uploaded as a *supporting information* file)A clean copy of the edited manuscript (uploaded as the new *manuscript* file

Additional Editor Comments (if provided):

Reviewers' comments:

Reviewer's Responses to Questions

**Comments to the Author**

1. Is the manuscript technically sound, and do the data support the conclusions?

Reviewer #1: Partly

Reviewer #2: Yes

Reviewer #3: Yes

Reviewer #4: No

2. Has the statistical analysis been performed appropriately and rigorously? 

Reviewer #1: No

Reviewer #2: Yes

Reviewer #3: Yes

Reviewer #4: No

3. Have the authors made all data underlying the findings in their manuscript fully available?

Reviewer #1: Yes

Reviewer #2: Yes

Reviewer #3: Yes

Reviewer #4: Yes

4. Is the manuscript presented in an intelligible fashion and written in standard English?

Reviewer #1: No

Reviewer #2: Yes

Reviewer #3: Yes

Reviewer #4: No

5. Review Comments to the Author

Reviewer #1: Morphological and molecular evidence for range extension and first occurrence of the Japanese seahorse, Hippocampus mohnikei (Bleeker 1853) in a bay-estuarine system of Goa, central west coast of India

The authors report of a seahorse, Hippocampus mohnikei usually found in the east Indian - and Pacific ocean, recorded for the first time on the west coast of India.

Overall, the study is interesting in its findings of range expansion of a small seahorse.

General comments:

I think the Authors did not read the Guidelines for submission?

Some sections are messy with Table captions missing or wrongly placed, Fig captions interrupting main text of MS … It was hard to follow and I put it away many times and nearly did not review!

Someone needs to check if all uploaded files are correct before sending to Reviewers!

Regarding the manuscript I would urge the authors to separate Tables and Figures with their captions from the main text of the MS. Figure legends/captions occurred within the text and disrupted the flow. Example: Table 2 starts on page 13 without Table legend/caption followed by text (new results). The actual table was (I assume) on page 14 without a header and table legend was below the table!

Another point for the morphological measurements; how about indicating the different measurements on one of the images of the seahorse with abbreviations (explained in Fig caption) e.g. see Forsgren & Lowe 2006 (seadragon morphology). Table2 would be more compact.

For the molecular work did you extract and sequence mitochondrial DNA (mtDNA) from all 4 individuals found or just from 2 individuals? I thought you caught 4? If you have 4 samples sequenced (COI and cytb) you should show all 4 in the tree not just the one. Where do the other 3 individuals occur in each tree (COI and cytb)? If all or some of the other 3 do not cluster with the voucher H. mohnikei then you have to consider other possibilities.

Some sections need to be deleted especially in Results! Result section is littered with sentences that end in eg ‘…. presented in Table x’ = not in published papers (more like student assignment)

Please explain your results and refer to your table/Fig in brackets!

Eg. Morphological measurements revealed ….. – state your results (Table 1).

Check grammar, punctuation in main text.

Specific comments:

Introduction

Line 57: Sentence; Often overlooked by most fishermen, due to their cryptic nature, their ability to camouflage and the sparse distribution restrict biologists to identify their presence in thick coastal marine habitats.

Line 100: Sentence about using gene markers re-write … sequenced 2 mtDNA gene regions (loci) COI & cyt b ..

Materials & Methods

Line126: incidental catch = is that bycatch - if so use correct term

Lines 132-141: Figure captions! Should be on separate sheet!

Line 175: ‘.. fin from fresh seahorse ..’ I thought they were dead seahorses from bycatch? Did I miss something?

Line 203: Seq of Goa specimens GB MK330041.1 and MK112274.2 = 2 seq’s submitted - did you not seq 4 individuals? If the other 2 did not provide DNA- seq (wwere they degraded?) you need to say this in your M&M!

add a table S1 (suppl) of the sequences (species name, GB accession # Ref) you downloaded for your phylogenetic tree from GB and cite their work.

Results

Line 236-237: delete senetence! Start explaining your main result and refer to your (Table/Fig) in brackets. ‘Morphological measurements revealed …Goa specimens are consistent with … (Table1).

I would move Table 1 into Suppl material (raw data).

Then describe some of the more important measurements that support your work.

Please don’t repeat results in your text if they are already in Table 1! People can read tables. So explain the measurements eg. on average size of … (mean #) comparable to size of mohnikei samples … from eg. east India or Thailand/Vietnam?

Line 258-259: re-write sentence … we compared … commonly occurring seahorses, H. kuda … (Table 2).

Pages 11-12, and 13-14 totally messy with tables throughout text making it difficult to follow should have been on separate sheet.

Page 13-14 = messy table is split I assume? No captions above table header - I assume Table 2? AND Table caption go ABOVE a table not below please read PlosOne Author Guidelines!

Table 3 - PCA – move to Suppl material it is already represented in Fig 4 – don’t repeat results.

Phylogenetic analysis – needs overhaul

Both trees (Fig5 & 6) only have 1 goa individual in tree. Where are the other 3 sequences?

You need to add all 4 indiv seq to each gene region (COI, cyt b) tree. If by chance other individuals did not fall into same clade as H mohnikei – this needs to be discussed! There are probably other reasons for it … (eg. hybrids?)

When you aligned the seq to the other GB downloaded seq in MEGA it should be obvious especially COI (Barcode of Life – check out H mohnikei on BOLD systems http://www.boldsystems.org/index.php/TaxBrowser_TaxonPage?taxid=66243)

Line 317 – delete ‘depicted in Fig5 and Fig6… see above comments

Line 325 – delete presented in Table …

Lines 330-338 = remove Fig captions from main MS text – refer to guidelines

Table 4 (don’t just copy table from program) = Table Header replace 1 2 3 4 with actual locations: Goa Japan …

Same with Table 5.

Line 317-372 Fig caption see guidelines

Discussion

Line 390-394 Fig caption – see guidelines

There are repeated results in the discussion – delete them – you already have these in your Result section. Discuss your results.

Line 405-415 remove results repeated – already in Result section

Figure 2 and Fig 3 = combine or place one in suppl material

Figure 1 & 8 combine = Fig 1 could be inset to the bigger Map Fig 8

Reviewer #2: Really nice description of morphological and molecular characters for the range expansion of H. mohnikei in India. This is a really interesting seahorse that requires more studies on its distributional range within the Indo-Pacific region.

Reviewer #3: General Comments:

This paper represents a significant contribution to the global knowledge of seahorse diversity in the Arabian Sea, and represents novel information that is interesting from both evolutionary and conservation perspectives. I suggest that the authors revise the manuscript to have more of a focus on the biogeography of seahorses in Southeast Asia, which is outlined nicely by Lourie and Vincent (2004). The manuscript is important and should be published, but requires a thorough re-write to correct the grammar (see specific comments below). The discussion needs to more clearly outline that this is likely to be an established population, but with the possibility that these specimens are vagrants.

The information regarding ocean circulation should be provided in the methods section, and figures should be added to display other seasons as well. This should then be linked in the discussion to how the species got to Goa. An effort should be made to determine where this species originates biogeographically and to clarify in which direction it dispersed over time.

Overall, this manuscript requires a substantial amount of work to be made publishable, but I do not consider this a 'major revision', as the overall structure and content need to change very little. I suggest publishing this manuscript in PLOS ONE after these extensive minor revisions have been made.

Lourie, S.A. and Vincent, A.C., 2004. A marine fish follows Wallace's Line: the phylogeography of the three‐spot seahorse (Hippocampus trimaculatus, Syngnathidae, Teleostei) in Southeast Asia. Journal of Biogeography, 31(12), pp.1975-1985.

Specific comments:

Line 27: use 'geographic range' rather than 'distribution range'; do this throughout the manuscript

Line 40: "hitherto known range of Japan and Vietnam" is incorrect. Aylesworth et al (2016) have already reported this species from the Gulf of Thailand and western Thailand.

Line 43: 'conspecificity' refers to individuals within the same species; it appears here that the authors are suggesting that the specimens from Japan are of a different species. I suggest rewording to state that the specimens examined in this study are closer genetically to those in Vietnam and Thailand than they are to those in Japan. Unless there is evidence to the contrary, they are all conspecifics.

Lines 56-57: 'unnoticeable from coastal habitat' is unwieldy. I suggest deleting this sentence.

Lines 57-59: This is an incomplete sentence that lacks a subject; 'thick coastal marine habitats' is ambiguous. I suggest deleting this sentence as well.

Line 60: at least three species have been described since Lourie et al. (2016): Hippocampus casscio, H. haema, and H. japapigu

Lines 63-64: the validity of H. montebelloensis and H. borboniensis is disptued by Lourie et al. (2016). Although it is not necessary to agree with this, it is necessary to make note in the manuscript.

Lines 69-70: "(e.g. pelvic and caudal fins), high variability and overlapping of body proportions, colour (camouflage) and skin filaments" should read "(e.g. pelvic and caudal fins), and have high variability of and overlapping body proportions, colour (camouflage) and skin filaments"

Line 73: 'relationships' rather than 'relationship'

Lins 80-81: currently reads "However, instances of small migration of seahorses in search of proper habitat", should read "However, instances of limited migrations by seahorses in search of proper habitat,"

Lines 83-84: I would not consider oceanic current to be 'stepping stones'. Although currents facilitate dispersal, the real 'stepping stones' would be patches of habitat along the way.

Lines 88-89: reference [35] (Aylesworth et al. 2016) report this species from western Thailand, not "from southeastern India to Korea and Japan"

Lines 89-91: 'Previous reports' should include the Aylesworth et al (2016) observations from western Thailand. As stated currently, it appears the authors of this manuscript have extended the range from Vietnam to western India, when in fact the manuscript is only extending it from eastern India to western India. (see Figure 27 in Lourie et al. 2016).

Lines 90-92: this species is not listed as Vulnerable because of 'recent exploitations'; it is listed as Vulnerable because of a suspected reduction in population size of >30% over the past 10 years due to fisheries exploitation.

Line 92: the word 'vulnerable' should be capitalized, as it is a formal IUCN Red List Category.

Line 98: there needs to be a bridge sentence: the current text indicates that surveys found no new specimens and then goes straight into the fact that mroe specimens were found. The fact that the specimens examined were caught as bycatch in bag fisheries needs to be explained here, prior to stating the expansion.

Lines 101-102: how do we know that this is a recent expansion of the range? Presumably this species could have been present here all along without being detected.

Line 109: "The collection site of seahorses" - suggest rewording is "The locality at which the examined seahorse specimens were collected from"

Line 119: suggest deleting 'characterized by'

Line 157: replace 'three time counting' with 'triplicate counting'

Line 158: change 'were also taken in case of male specimens' to 'were measured in all male specimens'.

Line 230: remove 'almost'

Lines 232-233: Are there two cheek spines and two eye spines in total, or on each side of the head (so four)?

Table 1: header needs to be repeated at top of second page

Line 258: replace 'A comparative' with 'A comparison of'

Line 260: change 'was' to 'were'

Line 265: delete 'the'

Line 266: it appears the heading for Table 2 became separated from the rest of the table.

Line 266: replace 'distinguishes' with 'distinguish'

Line 337: replace 'seahorses' with 'seahorse'

Lines 359-370: Unless the authors actually collected field data regarding ocean circulation, this section should remain entirely in the methods section where the study site is described.

Lines 386-388: should read "In northeast Asia, H. mohnikei is reported from China and South Korea".

Line 397: change 'morpho-molecular' to 'morphological and moleculat'; delete 'is the most recent'

Line 398: delete 'distributional'

Line 405: delete 'of' after 'constructed'

Line 410: replace 'sequences' with 'sequence'

Line 421: the term 'disjunct' is misused here. I don't believe there is any evidence to suggest any disjunctions in the population.

Lines 421-423: this statement needs some nuance. Genetic differences within a species do not necessarily mean that cryptic species exist. These difference can mean that there are cryptic species, but the authors should state what a defensible threshold for species delineation might be (i.e. at what % difference would we likely consider them separate species?).

Line 425: replace 'on' with 'of'

Line 431: replace 'report' with 'data'

Line 451: again I am not aware of any evidence to say that the population is disjunct

Lines 474-475: replace 'attain a settlement phase (benthic)' with 'settle on the benthos'

Line 476: the authors need to provide evidence that the population is disjunct

Line 484: delete 'a'; replace 'promoting' with 'permit'

Line 490: delete 'as'

Lines 488-500: Is there evidence that this species originated in China/Japan? This section reads as though the species dispersed here recently. How can we be sure it didn't originate in India and disperse to the east? The prevailing currents are only shown for one season.

Line 506: change 'entail seahorses to flow downstream' to 'force seahorses downstream'

Line 518: delete 'inhabiting'

Line 520: it's not clear what the authors mean by 'translocated populations'.

Line 528: Translocation suggests that humans collected the seahorses elsewhere and relocated them to Goa. I don't think that is what this paper suggests - the evidence is clear that this is probably an extension of their natural range based on new information. I suggest deleting point (i). This sentence should state that these are either vagrants or a previously unreported established population.

Reviewer #4: Mr. Rayadurga and colleagues are to be commended for their efforts to understand the taxonomic identity and biogeography of a rare and previously poorly documented seahorse species occurring on the central west coast of India. As the authors state, seahorses globally are in decline, driven by their demand as an ingredient in traditional Chinese medicine as well as by degradation of their coastal marine habitat by human activities. The taxonomy and distribution of seahorses must be further clarified if seahorse populations are to persist in the face of these pressures.

While indeed it is intriguing to find and identify several specimens of H. mohnikei in the Goa region, unfortunately the methods presented in this manuscript do not support the conclusions the authors draw, as explained below. In addition to the disconnect between the analyses and their conclusions, the findings of this paper are more appropriate for a more specialized journal, such as Marine Biodiversity Records or Zootaxa, than for PLoS ONE. The manuscript also needs a thorough editing by a native English speaker for grammar and clarity of prose. For all of these reasons, as explained in more detail, I cannot recommend this manuscript be published in PLoS ONE.

While indeed it is likely that the 4 analyzed seahorse specimens are H. mohnikei, neither the morphological or the genetic analyses have been conducted in a way that support the authors conclusions.

For the morphometric analysis, it is essential to include one or preferably more vouchered specimens of H. mohnikei in the analysis to conclusively determine that the morphometric cluster analysis supports the identification of the specimens in question. By examining only the 4 specimens in question against specimens from two other distinct seahorse species, the authors have only demonstrated that the unidentified seahorse specimens are NOT H. kuda or H. trimaculatus. But this analysis cannot be used to conclude these unidentified specimens ARE H. mohnikei. For that, you would need to also measure one or more vouchered specimens of H. mohnikei and show that the Goa specimens cluster together with known H. mohnikei morphometric data.

The case with the genetic data is the same. The correct analysis would be to download ALL VOUCHERED H. mohnikei COI and cytb sequences, align them, conduct an analysis of the correct model of molecular evolution for this dataset (there is no justification for why the authors used K2P - how was it determined that this was indeed the correct evolutionary model for these data?), and conduct a rigorous phylogenetic analysis that assesses the position of the 4 Goa specimens among all other known H. mohnikei sequences for both genes.

By choosing only a few sequences from other H. mohnikei specimens to compare to, the authors are depriving this analysis of the dataset required to more fully understand the identity and phylogeographic relationships of the Goa specimens.

The results of the genetic analysis do not support the conclusions the authors make, but potentially there is something quite interesting to be investigated here. The authors claim that the taxonomic identity of the Goa specimens in confirmed because the sequences of the putative H. mohnikei from Goa are nearly identical to just 3 other H. mohnikei sequences from southeast Asia. And here is where it gets even more interesting, while also pointing to the lack of support for the authors conclusions. The pairwise genetic distances reported between the Goa specimens and the 3 SE Asia sequences for both Cytb and COI are ENORMOUS, and absolutely cannot reflect intraspecific genetic variation. The authors are reporting 3-10% distance in COI, where most valid species can be separated at the 3% level. The cytb differences are so large as to reflect saturation in mutations - you basically cannot get larger genetic distances than 18% or so, and levels that high almost always reflect completely different species. These results only validate the need to include all known vouchered specimens of H. mohnikei in the analysis, to better understand the clustering among the 100 or so available specimens in GenBank and the Barcode of Life databases.

There are several possible reasons for these findings: an inaccurate sequence alignment, the need for a more sophisticated model of molecular evolution that can account for the possibility of multiple mutations at a given site, or that the specimens from Goa really are so genetically distinct they have diverged significantly on their evolutionary pathway. A more rigorous genetic analysis is the only way to understand what these 4 mystery specimens really represent.

I recommend the authors shorten their paper to a scientific note format and publish this work in a more specialized journal such as those listed above. By publishing the sequences and placing them in the public record, they will have done a great service to adding important information regarding the continuing mystery of seahorse taxonomy and distributions, for which they will be acknowledged.

6. PLOS authors have the option to publish the peer review history of their article (what does this mean?). If published, this will include your full peer review and any attached files.

Reviewer #1: No

Reviewer #2: Yes: Graham Short

Reviewer #3: Yes: Riley Pollom

Reviewer #4: No

---

## [Author Response · Author response to Decision Letter 0]

9 Jan 2020

Response to Comments/suggestions made by the Reviewers

Manuscript ID number: PONE-D-19-19415

Manuscript title: Morphological and molecular evidence for range extension and first occurrence of the Japanese seahorse, Hippocampus mohnikei (Bleeker 1853) in a bay-estuarine system of Goa, central west coast of India

Authors: Sushant Sanaye, Rakhee Khandeparker, Rayadurga Anantha Sreepada, Mamatha Singanahalli Shivaramu, Harshada Kankonkar, Jayu Narvekar, Mukund Gauthankar

We are thankful for the reviewers for their constructive criticisms/suggestions and their input has substantially improved the quality of the manuscript. 

Below, we provide a point-by-point response as to how the feedback received from the Reviewers has been incorporated in the revised manuscript.

This manuscript is a resubmission of manuscript PONE-D-19-19415 with our response to the questions made by the reviewers. The Academic Editor handling this manuscript is Dr. Rui Rosa

Reviewer #1: 

Morphological and molecular evidence for range extension and first occurrence of the Japanese seahorse, Hippocampus mohnikei (Bleeker 1853) in a bay-estuarine system of Goa, central west coast of India

The authors report of a seahorse, Hippocampus mohnikei usually found in the east Indian and Pacific ocean, recorded for the first time on the west coast of India. Overall, the study is interesting in its findings of range expansion of a small seahorse.

Response: We thank the Reviewer for the excellent review and the positive comments on the manuscript. Constructive suggestions provided by the Reviewer are appreciated.

General comments:

Comment: I think the Authors did not read the Guidelines for submission?. Some sections are messy with Table captions missing or wrongly placed, Fig captions interrupting main text of MS. It was hard to follow and I put it away many times and nearly did not review!. Someone needs to check if all uploaded files are correct before sending to Reviewers. Regarding the manuscript I would urge the authors to separate Tables and Figures with their captions from the main text of the MS. Figure legends/captions occurred within the text and disrupted the flow. Example: Table 2 starts on page 13 without Table legend/caption followed by text (new results). The actual table was (I assume) on page 14 without a header and table legend was below the table!

Response: At the outset, we apologise for the inconvenience caused due to the formatting issues. Here we state that the all the guidelines/instructions for online submission of the manuscript were meticulously adhered to. According to journal requirement, we have again checked our revised manuscript to meet PLOS ONE’s style requirements. We hope that the revised version will be the error-free readable format. 

Comment: Another point for the morphological measurements; how about indicating the different measurements on one of the images of the seahorse with abbreviations (explained in Fig caption) e.g. see Forsgren & Lowe 2006 (seadragon morphology). Table2 would be more compact.

Response: Highly appreciate constructive suggestion of the Reviewer. The measurement of morphological characters and meristic counts of seahorse specimens were carried out as per the standard protocols/procedures of measuring seahorses as outlined in Lourie et al. (1999); Lourie et al (2003) and Lourie et al (2004). These are standard protocols that have been used widely for the purpose. measurement of morphological and meristic characters of seahorse species by many authors. These references have been cited in sub-section, morphological analysis under Section, ‘Materials & Methods' which we consider serve the purpose. 

Comment: For the molecular work did you extract and sequence mitochondrial DNA (mtDNA) from all 4 individuals found or just from 2 individuals? I thought you caught 4? If you have 4 samples sequenced (COI and cytb) you should show all 4 in the tree not just the one. Where do the other 3 individuals occur in each tree (COI and cytb)? If all or some of the other 3 do not cluster with the voucher H. mohnikei then you have to consider other possibilities.

Response: We strongly appreciate the criticism provided by the Reviewer. As suggested by the reviewer, remaining 3 seahorse specimens (MK1, MK2 and MK3) were also sequenced in addition to originally sequenced seahorse specimen (MK4) for both COI and Cyt b genes. Sequences generated from 04 seahorse specimens with GenBank accession numbers, MN595216, MN595218, MN595217 and MK330041.1 for COI gene and MN595213, MN595214, MN595215 and MK112274.2 for Cyt b genes, respectively corresponding to 4 seahorse specimens (MK1, MK2, MK3 and MK4) were included in phylogenetic analysis (Fig. 6 and Fig. 7). Phylogenetic analysis revealed that all 4 seahorse individuals from Goa clustered together confirming a monophyletic lineage. Sequences generated of COI and Cyt b gene from 4 seahorse individuals have been used for phylogenetic analysis in the revised manuscript. 

Comment: Some sections need to be deleted especially in Results! Result section is littered with sentences that end in eg ‘…. presented in Table x’ = not in published papers (more like student assignment). Please explain your results and refer to your table/Fig in brackets! Eg. Morphological measurements revealed ….. state your results (Table 1). Check grammar, punctuation in main text.

Response: We have greatly modified the Results section as suggested by the Reviewer. The section of Results has been modified in the revised manuscript. We incorporated all grammatical and typographical edits into the revised text. The English language has been polished after incorporating the suggested grammatical and typographical edits into the text. Now, the English language of the manuscript has been checked by Grammarly software. 

Specific comments:

Introduction

Query: Line 57: Sentence; Often overlooked by most fishermen, due to their cryptic nature, their ability to camouflage and the sparse distribution restrict biologists to identify their presence in thick coastal marine habitats.

Response: As suggested by the reviewer, the sentence has been corrected. 

 

Query: Line 100: Sentence about using gene markers re-write … sequenced 2 mtDNA gene regions (loci) COI & cyt b ..

Response: The sentence corrected as suggested by the Reviewer.

Materials & Methods

Query: Line126: incidental catch = is that by catch - if so use correct term

Response: Correct terms used. In the revised manuscript, the sentence corrected as Dead seahorses landed as incidental catch in bag nets attached to the stakes operated by local fishermen ..............

Query: Lines 132-141: Figure captions! Should be on separate sheet!

Response: The MS has been prepared as per journal’s prescribed format. Accordingly, the tentative probable positioning of the Figures with captions has been shown in the manuscript 

Query: Line 175: ‘.fin from fresh seahorse ..’ I thought they were dead seahorses from by catch? Did I miss something?

Response: Actually, we wanted to infer freshly dead seahorses here. In the revised manuscript, this has been corrected as freshly landed dead seahorses' to bring clarity. 

Query: Line 203: Seq of Goa specimens GB MK330041.1 and MK112274.2 = 2 seq’s submitted - did you not seq 4 individuals? If the other 2 did not provide DNA- seq (were, they degraded?) you need to say this in your M&M! add a Table S1 (suppl) of the sequences (species name, GB accession # Ref) you downloaded for your phylogenetic tree from GB and cite their work.

Response: We are extremely grateful to the Reviewer for pointing out this mistake. As explained above, remaining 3 individual seahorses (MK1, MK2, MK3) were sequenced subsequently following the same protocols as that was followed for initially sequenced seahorse individual i.e. MK4. Sequences generated from all 4 seahorse individuals, MK1, MK2, MK3 and M were submitted at NCBI vide GenBank accession numbers, MN595216, MN595218, MN595217 and MK330041.1 for COI gene and MN595213, MN595214, MN595215 and MK112274.2 for Cyt b genes, respectively were included in phylogenetic analysis (Fig. 6 and Fig. 7). 

As advised by the reviewer, all the downloaded sequences of H. mohnikei for COI and Cyt b genes available at GenBank along with their voucher Number and references used for constructing phylogenetic trees has been provided Supplementary S1 Table. 

Results

Query: Line 236-237: delete sentence! Start explaining your main result and refer to your (Table/Fig) in brackets. ‘Morphological measurements revealed …Goa specimens are consistent with … (Table1). I would move Table 1 into Suppl material (raw data). Then describe some of the more important measurements that support your work. Please don’t repeat results in your text if they are already in Table 1! People can read tables. So explain the measurements eg. on average size of … (mean #) comparable to size of mohnikei samples from eg. east India or Thailand/Vietnam? Line 258-259: re-write sentence … we compared … commonly occurring seahorses, H. kuda … (Table 2). 

Response: As advised by the reviewer, Line, 236-237 has been deleted. Now, the sub section on 'morphological analysis' under Results section has been modified to include only the salient findings. We have modified the Results section to a great extent and hope to have improved it by doing so. 

Firstly, we have re-arranged the few paragraphs to better mirror the presentation sequence of the results. Additionally, we have introduced Table 3 "Comparison of diagnostic morphological characters of all vouchered specimens of Hippocampus mohnikei” for comprehensiveness of the morphological data. Since, Table 1 describes all the morphological characters of four seahorse specimens collected from Goa coast. Considering its significance to the manuscript, Table 1 has retained and not moved to Supplementary material. All other typos and corrections suggested by the Reviewer have been incorporated in the revised manuscript.

Query: Pages 11-12, and 13-14 totally messy with tables throughout text making it difficult to follow should have been on separate sheet.

Response: We apologise for the inconvenience caused due to formatting issues. The revised manuscript has been checked to meet PLOS ONE’s style requirements.

Query : Page 13-14 = messy table is split I assume? No captions above table header - I assume Table 2? AND Table caption go ABOVE a table not below please read Plos One Author Guidelines!

Response: We apologise for the inconvenience caused due to formatting issues. Now, the revised manuscript has been checked to meet PLOS ONE’s style requirements and hope the formatting issue is resolved.

Query : Table 3 - PCA – move to Suppl material it is already represented in Fig 4 – don’t repeat results.

Response: As suggested by the Reviewer, Table 3 on PCA output has moved to Supplementary material (S2 Table)

Query: Phylogenetic analysis – needs overhaul. Both trees (Fig. 5 & 6) only have 1 goa individual in tree. Where are the other 3 sequences? You need to add all 4 indiv seq to each gene region (COI, cyt b) tree. If by chance other individuals did not fall into same clade as H mohnikei – this needs to be discussed! There are probably other reasons for it …(eg. hybrids?). When you aligned the seq to the other GB downloaded seq in MEGA it should be obvious especially COI (Barcode of Life – check out H mohnikei on BOLD systems http://www.boldsystems.org/index.php/TaxBrowser_TaxonPage? taxid =66243)

Response: The subsection of 'Phylogenetic analysis' has been modified to a great extent by incorporating the sequence data of all 4 individual seahorses from Goa. Maximum likelihood (ML) trees for both COI and Cyt b genes have been constructed by aligning them with the downloaded sequences for all vouchered H. mohnikei downloaded from the GenBank in the revised manuscript. Phylogenetic analysis revealed that all 4 seahorse individuals from Goa clustered together and formed a single clade thus confirming a monophyletic lineage. This has been now mentioned in the revised manuscript. As advised by the reviewer, availability of sequences of COI gene for H. mohnikei were also checked in Barcode of Life (BOLD Systems). 

Query: Line 317 – delete ‘depicted in Fig. 5 and Fig. 6… see above comments

Response: Deleted

Query: Line 325 – delete presented in Table 

Response: Deleted 

Query: Lines 330-338 = remove Fig captions from main MS text – refer to guidelines.

Response: Figure captions removed from the main MS text. 

Query: Table 4 (don’t just copy table from program) = Table Header replace 1 2 3 4 with actual locations: Goa Japan Same with Table 5.

Response: As suggested by the reviewer, the Tables of genetic distance (Tables 4 & 5) have been reformatted in the revised manuscript.

Query: Line 317-372 Fig caption see guidelines.

Response: Done

Discussion

Query: Line 390-394 Fig caption – see guidelines

Response: Done

Query: There are repeated results in the discussion – delete them – you already have these in your Result section. Discuss your results.

Response: As suggested by the Reviewer, results have been deleted from the Discussion Section.

Query: Line 405-415 remove results repeated – already in Result section Fig. 2 and Fig 3 = combine or place one in suppl material Figure 1 & 8 combine = Fig 1 could be inset to the bigger Map Fig 8. 

Response: Following the comments made by the Reviewer, we have modified the discussion comprehensively. in the revised manuscript. Now, it is hoped the Discussion has improved by removing the repetition of Results. We tried to combine the Fig. 2 and Fig. 3, the quality of the Figure was lost. Similarly, we tried to integrate Fig. 1 to bigger map of Figure 8, but it did not work properly in terms of picture quality and alignment. 

Reviewer #2: 

Comment: Really nice description of morphological and molecular characters for the range expansion of H. mohnikei in India. This is a really interesting seahorse that requires more studies on its distributional range within the Indo-Pacific region.

Response: We thank the Reviewer for the excellent review with positive comments and appreciation of our work. We agree with the Reviewer that further studies on the distributional range of H. mohnikei within Indo-Pacific region in collaboration with other researchers in Southeast Asia would help in better understanding its evolutionary and biogeography.

Reviewer #3: 

General Comments

Comment: This paper represents a significant contribution to the global knowledge of seahorse diversity in the Arabian Sea, and represents novel information that is interesting from both evolutionary and conservation perspectives. I suggest that the authors revise the manuscript to have more of a focus on the biogeography of seahorses in Southeast Asia, which is outlined nicely by Lourie and Vincent (2004). [Lourie, S.A. and Vincent, A.C., 2004]. A marine fish follows Wallace's Line: the phylogeography of the three‐spot seahorse (Hippocampus trimaculatus, Syngnathidae, Teleostei) in Southeast Asia. Journal of Biogeography, 31(12), pp.1975-1985]. The manuscript is important and should be published, but requires a thorough re-write to correct the grammar (see specific comments below). The discussion needs to more clearly outline that this is likely to be an established population, but with the possibility that these specimens are vagrants.

Response: We gratefully thank the Reviewer for excellent review with positive comments/suggestions on the manuscript. As suggested by the Reviewer, the biogeography of seahorse populations in southeast Asia has been focussed in the revised manuscript. The paper by Lourie & Vincent (2004) dealing with the phylogeography of three-spotted seahorse, H. trimaculatus has been referred in the manuscript. Accordingly, the discussion on biogeography of seahorse populations has now been included in the discussion. Additional minor modifications have been made to improve overall readability and to clarify certain statements. The grammatical errors have also been fixed as per the specific comments/edits provided by the Reviewer.

The reviewer may agree that more samples of H. mohnikei would be needed to arrive at a definite conclusion for ascertaining whether the present sighting is a previously unreported established populations or vagrants. Based on this present reporting, our efforts for more specimens of H. mohnikei are still being pursued. With more number of specimens from Goa waters, the biogeography of this small seahorse species in southeast Asia could be addressed comprehensively. Of course this would help in better understanding from both evolutionary and conservation view points.

Comment: The information regarding ocean circulation should be provided in the methods section, and figures should be added to display other seasons as well. This should then be linked in the discussion to how the species got to Goa. An effort should be made to determine where this species originates biogeographically and to clarify in which direction it dispersed over time.

Response: As suggested by the reviewer, now the Figure describing the ocean circulation has been moved to Materials & Methods section (Fig. 4). The role of the ocean circulation in the Indo-Pacific region in facilitating the dispersal of H. mohnikei from its previously reported geographical range into the coastal waters of Goa has been now discussed in the revised manuscript.

Comment: Overall, this manuscript requires a substantial amount of work to be made publishable, but I do not consider this a 'major revision', as the overall structure and content need to change very little. I suggest publishing this manuscript in PLoS ONE after these extensive minor revisions have been made.

Response: The authors are thankful to the Reviewer for positive feedback on the manuscript. Constructive suggestions provided by the Reviewer are appreciated. Now the manuscript has been revised following the minor revisions suggested by the Reviewer. 

 

Specific comments

Query: Line 27: use 'geographic range' rather than 'distribution range'; do this throughout the manuscript

Response: As suggested, ‘distributional range’ changed to ‘geographical range’ in the revised manuscript.

Query: Line 40: "hitherto known range of Japan and Vietnam" is incorrect. Aylesworth et al (2016) have already reported this species from the Gulf of Thailand and western Thailand.

Response: We honestly agree with the Reviewer’s comment. Now, the statement has been modified to accurately represent the fact in the revised manuscript. 

Query: Line 43: 'conspecificity' refers to individuals within the same species; it appears here that the authors are suggesting that the specimens from Japan are of a different species. I suggest rewording to state that the specimens examined in this study are closer genetically to those in Vietnam and Thailand than they are to those in Japan. Unless there is evidence to the contrary, they are all conspecifics.

Response: Yes, Agreed. As suggested by the Reviewer, the statement has been reworded to genetically closer in the revised manuscript. 

Query: Lines 56-57: 'unnoticeable from coastal habitat' is unwieldy. I suggest deleting this sentence.

Response: Sentence deleted as suggested.

Query: Lines 57-59: This is an incomplete sentence that lacks a subject; 'thick coastal marine habitats' is ambiguous. I suggest deleting this sentence as well.

Response: As suggested by the Reviewer, the sentence as been modified to convey the intended meaning.

Query: Line 60: at least three species have been described since Lourie et al. (2016): Hippocampus casscio, H. haema, and H. japapigu

Response: We totally agree with the Reviewer and we are sorry for missing out this. With the reporting of additional 3 seahorse species since Lourie et al (2016), the statement has been modified to include this. i.e. 44 instead of 41. Appropriate reference (IUCN Seahorse, Pipefish and Stickleback Specialist Group (2019). https://www.iucn-seahorse.org (downloaded on 01/11/2019) has also been cited. 

Query: Lines 63-64: the validity of H. montebelloensis and H. borboniensis is disputued by Lourie et al. (2016). Although it is not necessary to agree with this, it is necessary to make note in the manuscript. 

Response: We have noted the point raised by the Reviewer. Accordingly, the sentence has been modified by including a caveat. Added a sentence ‘The validity of H. bornoinensis and H. montebelloensis as separate species however, is currently under revision due to their synonymity with H. kuda and H. zebra, respectively’ (Lourie et al., 2016). 

Query: Lines 69-70: "(e.g. pelvic and caudal fins), high variability and overlapping of body proportions, colour (camouflage) and skin filaments" should read "(e.g. pelvic and caudal fins), and have high variability of and overlapping body proportions, colour (camouflage) and skin filaments.

Response: As suggested by the Reviewer, the sentence has been modified.

Query: Line 73: 'relationships' rather than 'relationship'

Response: As suggested, ‘relationship’ changed to ‘relationships

Query: Lines 80-81: currently reads "However, instances of small migration of seahorses in search of proper habitat", should read "However, instances of limited migrations by seahorses in search of proper habitat

Response: As suggested, the statement has been modified. 

Query: Lines 83-84: I would not consider oceanic current to be 'stepping stones'. Although currents facilitate dispersal, the real 'stepping stones' would be patches of habitat along the way.

Response: We agree with the Reviewer to a certain extent. Oceanic currents may facilitate long distance dispersal but indirectly facilitate stepping stone mode for short distance dispersal. Accordingly, the sentence has been altered. 

Query: Lines 88-89: reference [35] (Aylesworth et al. 2016) report this species from western Thailand, not "from southeastern India to Korea and Japan"

Response: We are thankful to the Reviewer for pointing out this mistake. The statement has been now corrected in the revised manuscript. 

Query: Lines 89-91: 'Previous reports' should include the Aylesworth et al (2016) observations from western Thailand. As stated currently, it appears the authors of this manuscript have extended the range from Vietnam to western India, when in fact the manuscript is only extending it from eastern India to western India. (see Figure 27 in Lourie et al. 2016).

Response: Agreed. As suggested by the Reviewer, the sentence has been corrected in the revised manuscript. Yes it is true that the reported new range in the study is from eastern India to western India. 

Query: Lines 90-92: this species is not listed as Vulnerable because of 'recent exploitations'; it is listed as Vulnerable because of a suspected reduction in population size of >30% over the past 10 years due to fisheries exploitation.

Response: We totally agree with the Reviewer’s comment. As suggested, now the statement has been modified to convey the correct rationale for listing H. mohnikei as ‘VULNERABLE’

Query: Line 92: the word 'vulnerable' should be capitalized, as it is a formal IUCN Red List Category.

Response: In order to comply with the format of IUCN, the word ‘Vulnerable’ changed to ‘VULNERABLE’

Query: Line 98: there needs to be a bridge sentence: the current text indicates that surveys found no new specimens and then goes straight into the fact that more specimens were found. The fact that the specimens examined were caught as bycatch in bag fisheries needs to be explained here, prior to stating the expansion.

Response: We thank the Reviewer in pointing out this error. In order to have a smooth flow, a bridge sentence indicating the specimens examined were based on the incidental catch has been inserted in the revised manuscript (Lines, 111-114).

Query: Lines 101-102: how do we know that this is a recent expansion of the range? Presumably this species could have been present here all along without being detected.

Response: While it is true that this species could have been here without being detected all these years, but we are the first ones to report its occurrence along the west coast of India evidenced with both morphological and molecular data. Previous report of its occurrence from Indian waters by Thangaraj and Lipton (2007) is from southeastern coast of India and is solely based on the morphological data of a single specimen. As mentioned in the manuscript, intense sampling from coastal waters of west coast of India and comparison of H. mohnikei populations from Southeast Asian regions are required for comprehensive biogeography of this small seahorse species in the Indo-Pacific Region. 

Query: Line 109: "The collection site of seahorses" - suggest rewording is "The locality at which the examined seahorse specimens were collected from"

Response: Corrected as suggested.

Query: Line 119: suggest deleting 'characterized by'

Response: Deleted

Query: Line 157: replace 'three time counting' with 'triplicate counting'

Response: As suggested, 'three time counting' changed to 'triplicate counting'

Query: Line 158: change 'were also taken in case of male specimens' to 'were measured in all male specimens'.

Response: Corrected as suggested by the Reviewer.

Query: Line 230: remove 'almost'

Response: Removed

Query: Lines 232-233: Are there two cheek spines and two eye spines in total, or on each side of the head (so four)?

Response: Two prominent cheek spines and double rounded spines below the eye (each on the either side of the head) were observed

Query: Table 1: header needs to be repeated at top of second page

Response: We are sorry for the formatting issue. The size of the Table 1 has now been adjusted so that all the data of the Table to appear on the singe page. 

Query: Line 258: replace 'A comparative' with 'A comparison of' Line 260: change 'was' to 'were'

Query: Line 265: delete 'the'

Reply: Deleted.

Query: Line 266: it appears the heading for Table 2 became separated from the rest of the table.

Response: We are sorry for this. Table 2 has been reformatted and positioned to so that the heading and the data to appear on the single page. 

Query: Line 266: replace 'distinguishes' with 'distinguish'

Query: Line 337: replace 'seahorses' with 'seahorse'

Response: ‘seahorses’ replaced with ‘seahorse

Query: Lines 359-370: Unless the authors actually collected field data regarding ocean circulation, this section should remain entirely in the methods section where the study site is described.

Response: As suggested by the Reviewer, the Figure (Fig. 8) describing the ocean circulation has been moved to Materials & Methods section (now, Fig. 4). As mentioned in the manuscript, the basin-wide prevailing winter circulation (December–February) in the Indo-Pacific region based on the daily data for the 10 years (2007 to 2017) has been prepared and presented. 

Query: Lines 386-388: should read "In northeast Asia, H. mohnikei is reported from China and South Korea".

Response: As suggested, the sentence has been corrected. 

Query: Line 397: change 'morpho-molecular' to 'morphological and molecular'; delete 'is the most recent'

Response: As suggested, ‘morpho-molecular’ changed to ‘molecular and molecular’. Deleted ‘is the most recent’ in the revised manuscript.

Query: Line 398: delete 'distributional'

Response: Deleted as suggested

Query: Line 405: delete 'of' after 'constructed'

Query: Line 410: replace 'sequences' with 'sequence'

Response: Replaced ‘sequence’ with ‘sequences’

Query: Line 421: the term 'disjunct' is misused here. I don't believe there is any evidence to suggest any disjunctions in the population.

Response: We totally agree to the Reviewer’s view. In the revised manuscript, the term ‘disjunct’ has been removed. 

Query: Lines 421-423: this statement needs some nuance. Genetic differences within a species do not necessarily mean that cryptic species exist. These difference can mean that there are cryptic species, but the authors should state what a defensible threshold for species delineation might be (i.e. at what % difference would we likely consider them separate species?).

Response: 

Query: Line 425: replace 'on' with 'of' Line 431: replace 'report' with 'data'

Response: As suggested by the Reviewer, necessary edits have been incorporated in the revised manuscript

Query: Line 451: again I am not aware of any evidence to say that the population is disjunct

Response: We agree with the Reviewer and thank for pointing out this mistake. Now, the term ‘disjunct’ replaced with ‘geographically disconnected’

Query: Lines 474-475: replace 'attain a settlement phase (benthic)' with 'settle on the benthos'

Query: Line 476: the authors need to provide evidence that the population is disjunct

Response: Now, the term ‘disjunct’ replaced with ‘geographically disconnected’

Query: Line 484: delete 'a'; replace 'promoting' with 'permit'

Response: Necessary edits have been done.

Query: Line 490: delete 'as'

Response: Deleted

Query: Lines 488-500: Is there evidence that this species originated in China/Japan? This section reads as though the species dispersed here recently. How can we be sure it didn't originate in India and disperse to the east? The prevailing currents are only shown for one season.

Response: We thank the Reviewer for this interesting suggestion. There are no published evidence that this species originated in China/Japan but initial reported type specimens were from Japan/Vietnam and in recent years sighted/reported from other locations, as well. According to Teske et al possibility of originating H. mohnikei in India and dispersal to east is cannot be overruled but required supportive genetic data which we planning with more specimens and control region sequencing. We can also mention about this feature studies in conclusion

Query: Line 506: change 'entail seahorses to flow downstream' to 'force seahorses downstream'

Response: As suggested, necessary corrections have been incorporated in the revised manuscript

Query: Line 518: delete 'inhabiting'

Response: Deleted ‘ínhabiting'

Query: Line 520: it's not clear what the authors mean by 'translocated populations'.

Response: We are sorry for the usage of incorrect term here. Now, 'translocated' replaced with ‘geographically disconnected’

Query: Line 528: Translocation suggests that humans collected the seahorses elsewhere and relocated them to Goa. I don't think that is what this paper suggests - the evidence is clear that this is probably an extension of their natural range based on new information. I suggest deleting point (i). This sentence should state that these are either vagrants or a previously unreported established population.

Response: We are grateful to the Reviewer for pointing out this mistake and incorrect usage of the term, ‘translocation’ here. Concurring to the Reviewer’s suggestion, the point (i) has been deleted in the revised manuscript. 

Reviewer #4: 

General Comments

Comment: Mr. Rayadurga and colleagues are to be commended for their efforts to understand the taxonomic identity and biogeography of a rare and previously poorly documented seahorse species occurring on the central west coast of India. As the authors state, seahorses globally are in decline, driven by their demand as an ingredient in traditional Chinese medicine as well as by degradation of their coastal marine habitat by human activities. The taxonomy and distribution of seahorses must be further clarified if seahorse populations are to persist in the face of these pressures.

Response: We thank the Reviewer for positive feedback on the manuscript.

Comment: While indeed it is intriguing to find and identify several specimens of H. mohnikei in the Goa region, unfortunately the methods presented in this manuscript do not support the conclusions the authors draw, as explained below. In addition to the disconnect between the analyses and their conclusions, the findings of this paper are more appropriate for a more specialized journal, such as Marine Biodiversity Records or Zootaxa, than for PLoS ONE. The manuscript also needs a thorough editing by a native English speaker for grammar and clarity of prose. For all of these reasons, as explained in more detail, I cannot recommend this manuscript be published in PLoS ONE.

Response: We appreciate the Reviewer for providing the constructive criticisms. Here, we provide the point by point rebuttal/clarifications sought by the reviewer. While we agree that one of the shortcomings could be small sample size, nevertheless, the limited dataset was robust enough to provide interesting insights into H. mohnikei—small and highly migratory seahorse species. As the Reviewer agree that it is quite hard to obtain sufficient number of seahorse specimens for the study due to their cryptic nature and low population density. The revised version of the manuscript is more organised and focussed after incorporating the suggestions and corrections made by all four reviewers and performing additional analysis. We chose PLoS ONE as a medium of publishing our results, firstly because of the multidisciplinary nature of the journal and secondly the 'Open Access Policy' of the journal—which would allow wider dissemination of the results, quickly. In addition, we have seen few similar papers of this nature published 

Comment: While indeed it is likely that the 4 analyzed seahorse specimens are H. mohnikei, neither the morphological or the genetic analyses have been conducted in a way that support the authors conclusions.

Response: We thank the reviewer for constructive feedback. Following the Reviewer's comment, we have made two major changes to the manuscript which are explained point by point below. 

Comment: For the morphometric analysis, it is essential to include one or preferably more vouchered specimens of H. mohnikei in the analysis to conclusively determine that the morphometric cluster analysis supports the identification of the specimens in question. By examining only the 4 specimens in question against specimens from two other distinct seahorse species, the authors have only demonstrated that the unidentified seahorse specimens are NOT H. kuda or H. trimaculatus. But this analysis cannot be used to conclude these unidentified specimens ARE H. mohnikei. For that, you would need to also measure one or more vouchered specimens of H. mohnikei and show that the Goa specimens cluster together with known H. mohnikei morphometric data.

Response: As suggested by the Reviewer, now a rigorous morphological and morphometric analysis has been conducted A comparative assessment of diagnostic morphological features and morphometric characters of Goa specimens with all vouchered specimens of Hippocampus mohnikei has been conducted. Results of comparative analysis has been tabulated under Table 3. This comparison demonstrated that the unidentified specimens from Goa are H. mohnikei as the most of the diagnostic morphological characters of Goa specimens matched with those of vouchered specimens. 

Comment: The case with the genetic data is the same. The correct analysis would be to download ALL VOUCHERED H. mohnikei COI and cytb sequences, align them, conduct an analysis of the correct model of molecular evolution for this dataset (there is no justification for why the authors used K2P - how was it determined that this was indeed the correct evolutionary model for these data?), and conduct a rigorous phylogenetic analysis that assesses the position of the 4 Goa specimens among all other known H. mohnikei sequences for both genes.

By choosing only a few sequences from other H. mohnikei specimens to compare to, the authors are depriving this analysis of the dataset required to more fully understand the identity and phylogeographic relationships of the Goa specimens.

Response: We thank the Reviewer for pointing out this mistake. As suggested by the Reviewer, we have downloaded all vouchered H. mohnikei COI and Cyt b sequences from GenBank (S1 Table), aligned them and conducted a rigorous phylogenetic analysis. This rigorous analysis allowed to assess the phylogeographic analysis of 4 specimens for Goa among all other known H. mohnikei sequences for both COI and Cyt b genes. 

Comment: The results of the genetic analysis do not support the conclusions the authors make, but potentially there is something quite interesting to be investigated here. The authors claim that the taxonomic identity of the Goa specimens in confirmed because the sequences of the putative H. mohnikei from Goa are nearly identical to just 3 other H. mohnikei sequences from southeast Asia. And here is where it gets even more interesting, while also pointing to the lack of support for the authors conclusions. The pairwise genetic distances reported between the Goa specimens and the 3 SE Asia sequences for both Cytb and COI are ENORMOUS, and absolutely cannot reflect intraspecific genetic variation. The authors are reporting 3-10% distance in COI, where most valid species can be separated at the 3% level. The cytb differences are so large as to reflect saturation in mutations - you basically cannot get larger genetic distances than 18% or so, and levels that high almost always reflect completely different species. These results only validate the need to include all known vouchered specimens of H. mohnikei in the analysis, to better understand the clustering among the 100 or so available specimens in GenBank and the Barcode of Life databases.

There are several possible reasons for these findings: an inaccurate sequence alignment, the need for a more sophisticated model of molecular evolution that can account for the possibility of multiple mutations at a given site, or that the specimens from Goa really are so genetically distinct they have diverged significantly on their evolutionary pathway. A more rigorous genetic analysis is the only way to understand what these 4 mystery specimens really represent.

Response: Post-sequencing of remaining 3 Goa specimens for COI and Cyt b genes (now, total four) and aligning them with all vouchered downloaded H. mohnikei COI and Cyt b sequences from GenBank, the results of phylogenetic analysis has improved greatly. The revised manuscript has addressed most of the the apprehensions, particularly on the genetic distance raised by the Reviewer. The manuscript has been now checked and verified by Grammarly software for English language. 

Comment: I recommend the authors shorten their paper to a scientific note format and publish this work in a more specialized journal such as those listed above. By publishing the sequences and placing them in the public record, they will have done a great service to adding important information regarding the continuing mystery of seahorse taxonomy and distributions, for which they will be acknowledged.

Response: We thank the Reviewer for this suggestion. Considering the sighting of H. mohnikei along the west coast of India has biogeographic and conservation significance, the manuscript attempts to address the probable reasons for its occurrence and range expansion. We wanted the manuscript to be quite different from routine Scientific Notes merely reporting the occurrence of this highly migratory small seahorse species. We hope this manuscript would provide new information and deeper insights which may be helpful in addressing the continuing mystery of seahorse taxonomy and distributions.

---

## [Decision Letter · Decision Letter 1]

27 Feb 2020

PONE-D-19-19415R1

Morphological and molecular evidence for range extension and first occurrence of the Japanese seahorse, Hippocampus mohnikei (Bleeker 1853) in a bay-estuarine system of Goa, central west coast of India

PLOS ONE

Dear Mr Rayadurga,

Thank you for submitting your manuscript to PLOS ONE. After careful consideration, we feel that it has merit but does not fully meet PLOS ONE’s publication criteria as it currently stands. Therefore, we invite you to submit a revised version of the manuscript that addresses the points raised during the review process.

We would appreciate receiving your revised manuscript by Apr 12 2020 11:59PM. To enhance the reproducibility of your results, we recommend that if applicable you deposit your laboratory protocols in protocols.io, where a protocol can be assigned its own identifier (DOI) such that it can be cited independently in the future. For instructions see: http://journals.plos.org/plosone/s/submission-guidelines#loc-laboratory-protocols

We look forward to receiving your revised manuscript.

Kind regards,

Rui Rosa

Academic Editor

PLOS ONE

Reviewers' comments:

Reviewer's Responses to Questions

**Comments to the Author**

1. If the authors have adequately addressed your comments raised in a previous round of review and you feel that this manuscript is now acceptable for publication, you may indicate that here to bypass the “Comments to the Author” section, enter your conflict of interest statement in the “Confidential to Editor” section, and submit your "Accept" recommendation.

Reviewer #1: All comments have been addressed

Reviewer #2: All comments have been addressed

Reviewer #3: (No Response)

2. Is the manuscript technically sound, and do the data support the conclusions?

Reviewer #1: Yes

Reviewer #2: Yes

Reviewer #3: Yes

3. Has the statistical analysis been performed appropriately and rigorously? 

Reviewer #1: Yes

Reviewer #2: Yes

Reviewer #3: Yes

4. Have the authors made all data underlying the findings in their manuscript fully available?

Reviewer #1: Yes

Reviewer #2: Yes

Reviewer #3: (No Response)

5. Is the manuscript presented in an intelligible fashion and written in standard English?

Reviewer #1: Yes

Reviewer #2: Yes

Reviewer #3: Yes

6. Review Comments to the Author

Reviewer #1: My previous comments have all been answered. I still have 2 minor edits, please see attached word file.

Reviewer #2: (No Response)

Reviewer #3: The authors have substantially improved the manuscript based on my previous suggestions. However, there is still a substantial amount of wordsmithing that is needed to make this manuscript publishable.

In particular, there is some confusion around active and passive dispersal (a one-time event) vs migration (which usually happens seasonally). I suggest the authors have a thorough read-through to ensure these terms are differentiated appropriately throughout the text, and particularly in the discussion and conclusions.

Please see the attached pdf with my edits and comments for suggested changes.

7. PLOS authors have the option to publish the peer review history of their article (what does this mean?). If published, this will include your full peer review and any attached files.

Reviewer #1: No

Reviewer #2: No

Reviewer #3: Yes: Riley A. Pollom

---

## [Author Response · Author response to Decision Letter 1]

2 Mar 2020

Response to Comments/suggestions made by the Reviewers

Manuscript ID number: PONE-D-19-19415R1

Manuscript title: Morphological and molecular evidence for range extension and first occurrence of the Japanese seahorse, Hippocampus mohnikei (Bleeker 1853) in a bay-estuarine system of Goa, central west coast of India

Authors: Sushant Sanaye, Rakhee Khandeparker, Rayadurga Anantha Sreepada, Mamatha Singanahalli Shivaramu, Harshada Kankonkar, Jayu Narvekar, Mukund Gauthankar

We are thankful for the reviewers for their constructive criticisms/suggestions on the revised version of the manuscript and their inputs have substantially improved the quality of the manuscript. 

Below, we provide a point-by-point response as to how the feedback received from the Reviewers (#1 and #3) has been incorporated in the revised manuscript.

This manuscript is a resubmission of manuscript PONE-D-19-19415R1 with our response to the questions made by the reviewers. The Academic Editor handling this manuscript is Dr. Rui Rosa

Reviewer #1: 

Reviewer #1: My previous comments have all been answered. I still have 2 minor edits. Please see attached word file.

Great improvement! So much easier to read and follow!. Only 2 minor edits:

Response: We thank the Reviewer for the excellent review and the positive comments on the manuscript. Constructive suggestions provided by the Reviewer are appreciated.

Minor Edit # 1

Lines 61-64 please see reviewers (#1 & #3) comments: delete sentence line 61-62

Lines 62-64 either delete sentence or change sentence it is still awkward. 

What you’re trying to say is that they are well camouflaged and rare and thus easily missed (not seen) by biologists/divers etc.

Response: As suggested by the Reviewer #1 and also by the Reviewer #3, the sentences, 61-64 have been modified in the revised manuscript as below:

However, these biological traits often also make them challenging for scientists to research and quantify. Due to their cryptic nature, their ability to camouflage and the sparse distribution, seahorses become almost unnoticeable and thus restricting the biologists for their identification in dense coastal marine habitats

Minor Edit # 2

Line 150= incidental catch = … landed as bycatch … is the correct word (eg. fish caught in nets/bags that was not intended)

Line 147 use the word bycatch throughout … incidentally caught as bycatch in bags …

Line 150 dead seahorses landed as bycatch …

Response: As suggested by the reviewer, the sentences under line numbers, 147-150 have been corrected as bycatch. However, there is a significant differences between countries between with regard to the interpretation of terms, bycatch’’ and ‘incidental catch’ exists. FAO’s explanatory notes on these terms can be found at the below link:

http://www.fao.org/3/y5936e/y5936e08.htm

Reviewer #3: 

The authors have substantially improved the manuscript based on my previous suggestions. However, there is still a substantial amount of wordsmithing that is needed to make this manuscript publishable. 

In particular, there is some confusion around active and passive dispersal (a one-time event) vs migration (which usually happens seasonally). I suggest the authors have a thorough read-through to ensure these terms are differentiated appropriately throughout the text, and particularly in the discussion and conclusions. 

Please see the attached pdf with my edits and comments for suggested changes.

Comments: Corrections to the Title of the manuscript

Response: As suggested by the Reviewer, the title has been modified to 'Morphological and molecular evidence for first records and range extension of the Japanese seahorse, Hippocampus mohnikei (Bleeker 1853) in a bay-estuarine system of Goa, central west coast of India

Response: We gratefully thank the Reviewer for excellent review with positive comments/suggestions on the manuscript. As suggested by the Reviewer, all the suggested edits indicated in the PDF have been incorporated in the revised manuscript. 

Here, we provide following responses to the specific queries pointed out by the Reviewer.

Comment: Shifting & deletion of Lines 106-114

Response: These lines form the background and main purpose of undertaking the present study. Therefore these lines retained in order to have to a flow and continuity in the Discussion also.

Comment: Why only the winter season? It needs to be explained why this data was only collected over one season. 

Response: The Indo-Pacific region encompasses the East Asian countries and Indian subcontinent comes under the influence of strong seasonal monsoon wind reversal and associated reversals in the surface currents. It is only during winter monsoon season, i.e. from November until February, the surface circulation in the north Indian Ocean including the Indo-Pacific region experiences currents pattern is from east to west which is shown in Fig. 4.

During summer monsoon season (June to September), the surface current in this region reverses and at this time of the year current flows from west to east. Although the prevailing currents during summer monsoon season was also examined but did not present as the circulation pattern during the summer monsoon cannot support a passive dispersal of marine fish species from east to west.

Due to lack of supportive genetic data from the coastal waters of India and from South East Asia region, the possibility of H. mohnikei originating in India and its dispersal to east was not considered in the present study. With the availability of genetic data from larger sample size and sequencing of mitochondrial DNA (mtDNA) gene, it may be possible to understand the evolutionary and biogeography of H. mohnikei in the Indo-Pacific region. This has been mentioned in the Conclusion section of the revised manuscript.

Now, the rationale for focusing on winter circulation has been added at the beginning of the sub-section Prevailing ocean circulation’ under the lines 260–268 in the revised manuscript. 

Comment: were genetic comparisons also made with these species from that area?

Response: Phylogenetic trees constructed with COI and Cyt b genes of H. mohinkei with H. kuda and H. trimaculatus separated out. Genetic sequence data of H. kuda and H. trimaculatus in fact were included in the pre-revised manuscript. But were replaced with a new set of phylogenetic trees (comparing the vouchered specimens) as suggested by one of the Reviewers. Since the morphological features and meristic characters of Goa specimens matched with that of vouchered specimens of H. mohnikei, a detailed genetic analysis of Goa specimens was performed for species reconfirmation and genetic placement of Goa specimens.

 

Comment: Line Nos. 351-353: combine all accession numbers in a single set of brackets.

Response: GenBank accession Numbers refer to sequences of COI and Cyt b genes. Combining all the sequences into a single bracket would be confusing. Hence the status quo retained. 

Comment: Figures 6 and 7 are too low-resolution and are not discernible

Response: The probable reason for reduction in the resolution may be due to reformatting issues. Converting PACE generated figures to PDF. New high resolution figures have been now uploaded along with the revised manuscript. 

Comment: Lines, 509-511: It's not clear to me why the high density of prey items would lead to a prolonged planktonic phase. Please elaborate.

Lines, 517-519- again, why the prolonged planktonic phase?

Response: The sentence has been modified/restructured to convey the intended meaning. Further, the reasoning for extension of planktonic phase by juveniles of H. mohnikei has been provided in the revised manuscript.

---

## [Editor Report · Decision Letter 2]

4 Mar 2020

Morphological and molecular evidence for first records and range extension of the Japanese seahorse, Hippocampus mohnikei (Bleeker 1853) in a bay-estuarine system of Goa, central west coast of India

PONE-D-19-19415R2

Dear Dr. Rayadurga,

We are pleased to inform you that your manuscript has been judged scientifically suitable for publication and will be formally accepted for publication once it complies with all outstanding technical requirements.

With kind regards,

Rui Rosa

Academic Editor

PLOS ONE
---

## [Editor Report · Acceptance letter]

9 Mar 2020

PONE-D-19-19415R2 

Morphological and molecular evidence for first records and range extension of the Japanese seahorse, *Hippocampus mohnikei* (Bleeker 1853) in a bay-estuarine system of Goa, central west coast of India 

Dear Dr. Rayadurga:

I am pleased to inform you that your manuscript has been deemed suitable for publication in PLOS ONE. Congratulations! Your manuscript is now with our production department. 

With kind regards,

on behalf of

Dr. Rui Rosa 

Academic Editor

PLOS ONE